# A quantitative analysis of the interplay of environment, neighborhood, and cell state in 3D spheroids

Vito RT Zanotelli[1,2], Matthias Leutenegger[3], Xiao-Kang Lun[2,3,5] iD, Fanny Georgi[2,3] iD, Natalie de Souza[1,4] & Bernd Bodenmiller[1,*] iD

## Abstract

Cells react to their microenvironment by integrating external stimuli into phenotypic decisions via an intracellular signaling network. To analyze the interplay of environment, local neighborhood, and internal cell state effects on phenotypic variability, we developed an experimental approach that enables multiplexed mass cytometric imaging analysis of up to 240 pooled spheroid microtissues. We quantified the contributions of environment, neighborhood, and intracellular state to marker variability in single cells of the spheroids. A linear model explained on average more than half of the variability of 34 markers across four cell lines and six growth conditions. The contributions of cell-intrinsic and environmental factors to marker variability are hierarchically interdependent, a finding that we propose has general implications for systems-level studies of single-cell phenotypic variability. By the overexpression of 51 signaling protein constructs in subsets of cells, we also identified proteins that have cell-intrinsic and cell-extrinsic effects. Our study deconvolves factors influencing cellular phenotype in a 3D tissue and provides a scalable experimental system, analytical principles, and rich multiplexed imaging datasets for future studies.

**Keywords** high-throughput assay; multiplexed imaging; spatial signaling; spatial variance; tissue organization
**Subject Categories** Cancer; Methods & Resources; Signal Transduction
**Mol Syst Biol. (2020) 16: e9798**

## Introduction

The ability of a cell to sense and adapt to its local environment depends on an intracellular signaling network that integrates paracrine, juxtacrine, nutritional, and mechanical cues to drive phenotypic decisions (Fig 1A). Genomic alterations that deregulate environment sensing and signaling can enable cells to grow outside their physiologically permissive tissue context, leading to diseases such as cancer. Since even strongly deregulated cells depend on and react to microenvironmental cues (Snijder & Pelkmans, 2011; Battich *et al*, 2015), microenvironment-induced cellular plasticity may contribute to the clinically relevant tumor cell heterogeneity observed in cancer tissues (Marusyk *et al*, 2012; Bodenmiller, 2016).

Assessments of spatial heterogeneity for several types of tumors have been performed based on protein and transcript measurements (Shah *et al*, 2017; Regev *et al*, 2017; Moffitt *et al*, 2018; Keren *et al*, 2018; Ali *et al*, 2020; Jackson *et al*, 2020; Schürch *et al*, 2020). Missing, however, is a quantitative understanding of how the tissue environment influences heterogeneity. Existing atlases of cancer tissues are based on static measurements of cellular markers that cannot reliably discriminate environment-dependent phenotypic plasticity from phenotypic variation due to genomic or lineage differences (Wagner *et al*, 2016; Regev *et al*, 2017). To quantify variability caused by the environment, it is necessary to identify comparable cells that vary phenotypically only because their environments differ.

To address this issue, we developed a system to quantitatively study multicellular spheroids consisting of clonal cells (Kunz-Schughart, 1999). We reasoned that this type of homogenous system would serve as a simplified model to quantify the influence of the global environment, local environment, and cell state on measurable cellular phenotypes. Further, as spheroid cell culture is compatible with 96-well microplates, this technology is suitable for large-scale perturbation studies and can be extended to more complex co-culture systems or heterocellular organoids (Friedrich *et al*, 2009; Wenzel *et al*, 2014; Fu *et al*, 2017; Qin *et al*, 2020).

To efficiently quantify phenotypic and signaling states of cells in spheroids at high throughput, we coupled metal-based barcoding

1 Department of Quantitative Biomedicine, University of Zurich, Zürich, Switzerland
2 Life Science Zürich Graduate School, ETH Zürich and University of Zürich, Zürich, Switzerland
3 Department of Molecular Life Sciences, University of Zurich, Zürich, Switzerland
4 Institute of Molecular Systems Biology, ETH Zurich, Zürich, Switzerland
5 Wyss Institute for Biologically Inspired Engineering, Harvard University, Boston, MA, USA
*Corresponding author. Tel: +41 44 635 31 28; E-mail: bernd.bodenmiller@imls.uzh.ch

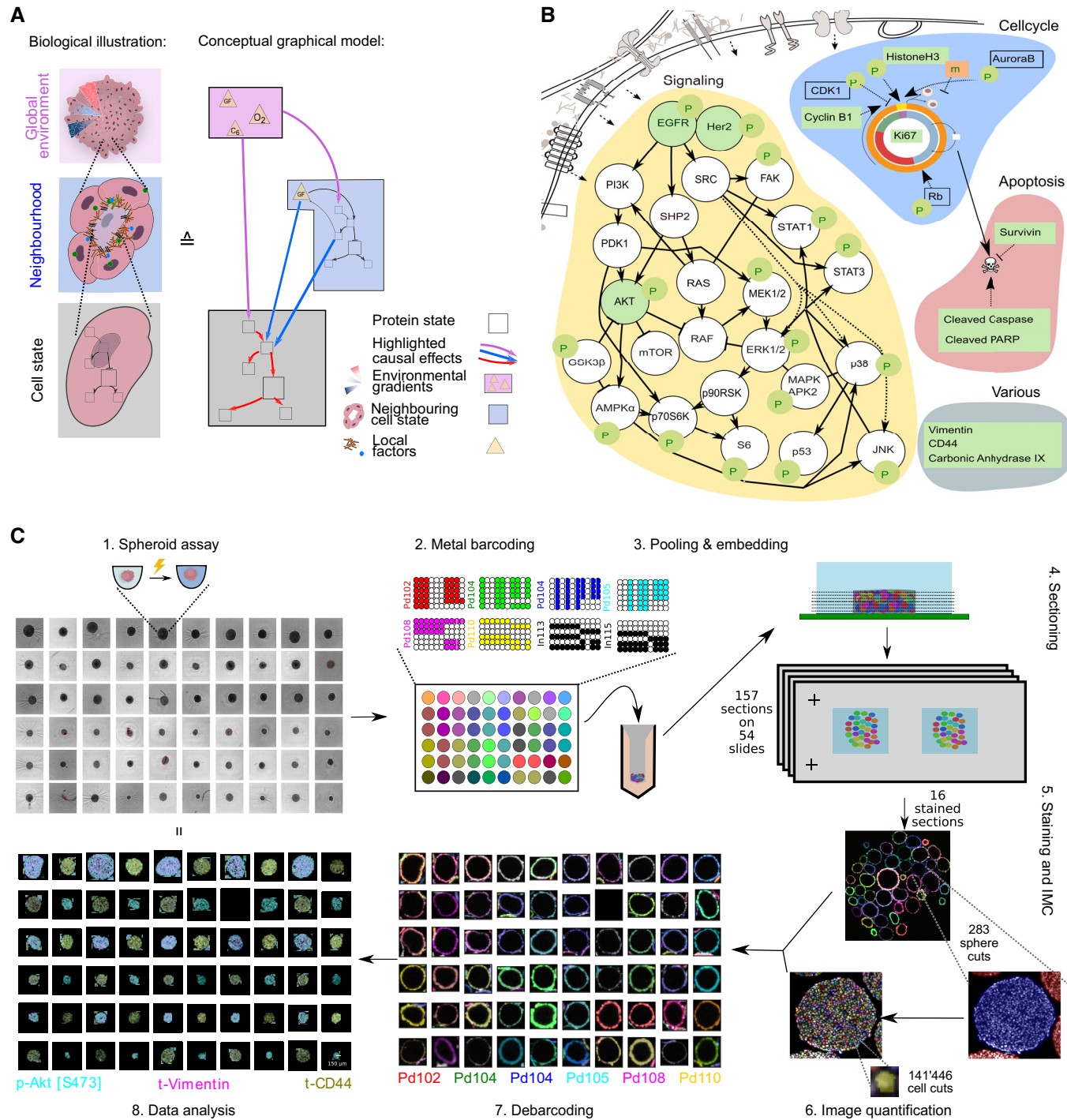

**Figure 1. Barcoded IMC assays allow efficient spatial profiling of pooled spheroids.**

A   Cells sense their environment and compute cellular decisions via a signaling network. Left: Depiction of spheroids at different scales: spheroid with global gradients, for example, of nutrients and oxygen (top), cellular neighborhood (middle), and single cell (bottom). Right: A schematic graphical model highlighting how global environment (pink box), local neighborhood (blue box), and intracellular state (gray box) can determine the levels of a given marker.

B   A schematic illustration of the signaling network markers, cell state markers, and other phenotypic markers measured using IMC (green). Nodes depicted in white were not measured.

C   Diagram of the approach used for multiplexed IMC analyses of spheroids. The image quantification step involves extraction of information in the form of tabular measurements from images. The data analysis step includes project-specific, statistical analyses of extracted measurements and their relationships to the different perturbations used.

with antibody-based multiplexed imaging mass cytometry (IMC) (Bodenmiller *et al*, 2012; Giesen *et al*, 2014; Zunder *et al*, 2015). This approach allowed us to process up to 240 spheroids simultaneously and to measure the levels of dozens of phenotypic markers in hundreds of sphere slices containing hundreds of thousands of cell sections. We evaluated spheroids formed by four cell lines, each grown in six different growth conditions, quantified single-cell marker levels, and analyzed how cell state, local neighborhood, and global environment interact to contribute to cell-to-cell variability in marker expression. Further, to explicitly probe cell-to-cell signaling interactions, we developed a chimeric overexpression-based approach to test the effects of overexpression of 51 ligand and receptor components of more than a dozen different signaling pathways on responses of neighboring cells. We observed that internal cell state and environmental features are strongly interdependent in their influence on marker variability, a finding that should be taken into account in systems-level studies of more heterogenous tissues as well. Our approach provides a blueprint for large-scale, multiplexed imaging studies on any 3D microtissue and for deconvoluting microenvironmental and internal contributions to cellular phenotype in spatial data.

# Results

### Spheroid culture coupled with multiplexed imaging enables quantification of phenotypic variability

To investigate factors that influence phenotypic variability in spheroids consisting of clonal cells, we developed a combined experimental and computational workflow. We grew cells as spheroids and imaged histological sections of these 3D tissues using IMC (Giesen *et al*, 2014). We used a panel of antibodies that detect 20 growth signaling markers, nine cell-cycle or apoptosis markers, and three markers capturing other molecular phenotypes (Fig 1B, Dataset EV1). We characterized the internal state of each cell by quantifying marker levels in individual cell sections; in this paper, cell state is defined as measurements of all intracellular marker levels (Fig 1A, gray box). We evaluated the local environment of a cell by quantifying marker levels within neighboring cells (Fig 1A, blue box). Finally, since this culture system shows radially symmetric gradients of nutrients, oxygen, and growth factors (Carlsson & Acker, 1988; Kunz-Schughart, 1999; Hirschhaeuser *et al*, 2010), we used an estimate of the distance from a cell to the border of the spheroid as a surrogate measurement of global environmental influences on phenotype (Fig 1A, violet box, Fig EV1C "Processing").

Histological sectioning, staining, and quantitative analysis of individual 3D microtissues are challenging to perform at scale: Cutting and staining spheres individually are very labor- and resource-intensive. To improve scalability, we adapted a metal-based barcoding approach from single-cell mass cytometry (Bodenmiller *et al*, 2012; Zunder *et al*, 2015) (Figs 1C and EV1A). This approach enabled barcoding of up to 240 single spheroids grown in individual wells of multi-well plates. After barcoding, spheres were pooled into a dense cylinder for efficient embedding and cutting. Sections from the spheroid plug were then imaged using bright-field imaging, and sections containing dozens of spheres were selected for staining and IMC analysis. The metal barcodes allowed us to

relate each imaged sphere section to its sphere of origin and thus to the cell line and perturbation (Fig EV2A–C). Pooled processing of spheres reduced the manual labor and processing variability, and staining of spatially concentrated spheres reduced the amount of antibody required compared with other approaches (Ivanov & Grabowska, 2017). Finally, we improved data quality by applying rigorous quality control steps on the cell, sphere slice, and intact sphere data by leveraging orthogonal imaging modalities such as bright-field and fluorescent imaging (Fig EV1B and C). Quantitative analysis on this scale necessitates thorough quality control to avoid technical artifacts.

We grew spheroids from four widely used epithelial cell lines that reproducibly form smooth spheroids (Zanoni *et al*, 2016). T-47D cells are derived from a breast cancer tumor (Holliday & Speirs, 2011), HT-29 and DLD-1 lines are derived from colorectal tumors (Dexter *et al*, 1981; Fogh, 2013), and T-REx-293 cells are derived from human embryonic kidney cells (Stepanenko & Dmitrenko, 2015). We chose these cell lines with the goal of identifying cell line-specific and general factors that influence phenotypic variability. In addition, to examine whether our results were affected by spheroid size or growth time, we grew each of these four cell lines at three cell seeding concentrations (5 replicate wells each) and for two different time periods (72 and 96 h) resulting in a total of 120 spheroids (Fig EV2D). After cutting the pooled spheroid pellets, sections were stained with our antibody panel (Fig 1B, Dataset EV1) and imaged using IMC. After quality control and image processing, our data included 517 cuts from 100 spheres, corresponding to 228,740 cell sections with an average of 19,530 cell sections per cell line and growth condition (min = 1,426, max = 28,170, Dataset EV2). This corresponded to an average of 5 randomly selected sections per sphere.

### Marker levels show strong dependence on environment and are cell intrinsically and spatially correlated

We segmented the imaged spheroids into single-cell sections using a combination of machine learning and computer vision algorithms and quantified the average level of each measured marker for each single cell. Dimensionality reduction analysis showed a near-perfect separation into cells of the different cell lines as identified by debarcoding (Fig EV3A). We further confirmed that there were likely no misassignments during debarcoding with a clustering-based analysis (Fig EV3B–D). Visual inspection of spheroid images showed clear marker-specific spatial variation. Certain markers appeared in patches of cells, whereas the levels of other markers were dependent on distance to the spheroid border (Fig 2A and B). These results indicate that both the local environment and global effects influence marker expression.

To systematically investigate intracellular, local, and global relationships for the 34 markers measured, we calculated Pearson's correlations between intracellular levels of each marker in a given cell (cell state) and between intracellular markers and the average levels of markers in the immediate neighbors of the cell (local neighborhood) (Fig 2C). We also calculated the distance from the cell to the spheroid border as a proxy for the global environment and visualized average marker levels relative to this distance. The results for the HT-29 cell line are representative (see Fig EV4A–C, for example, data on all cell lines), and analyses of this cell line in one growth condition are discussed in this section.

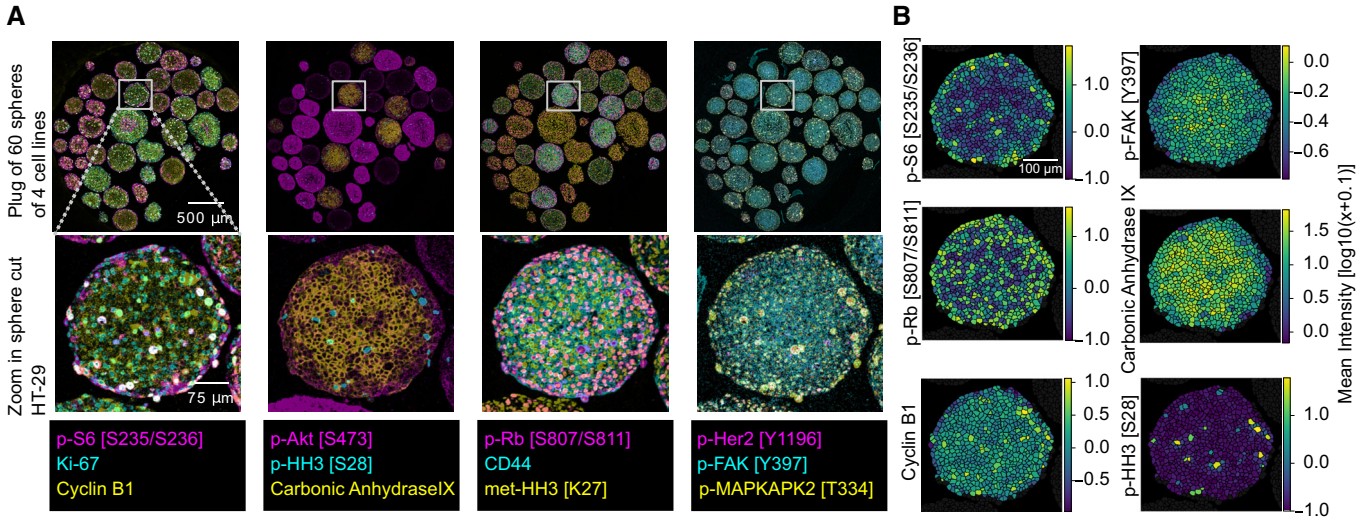

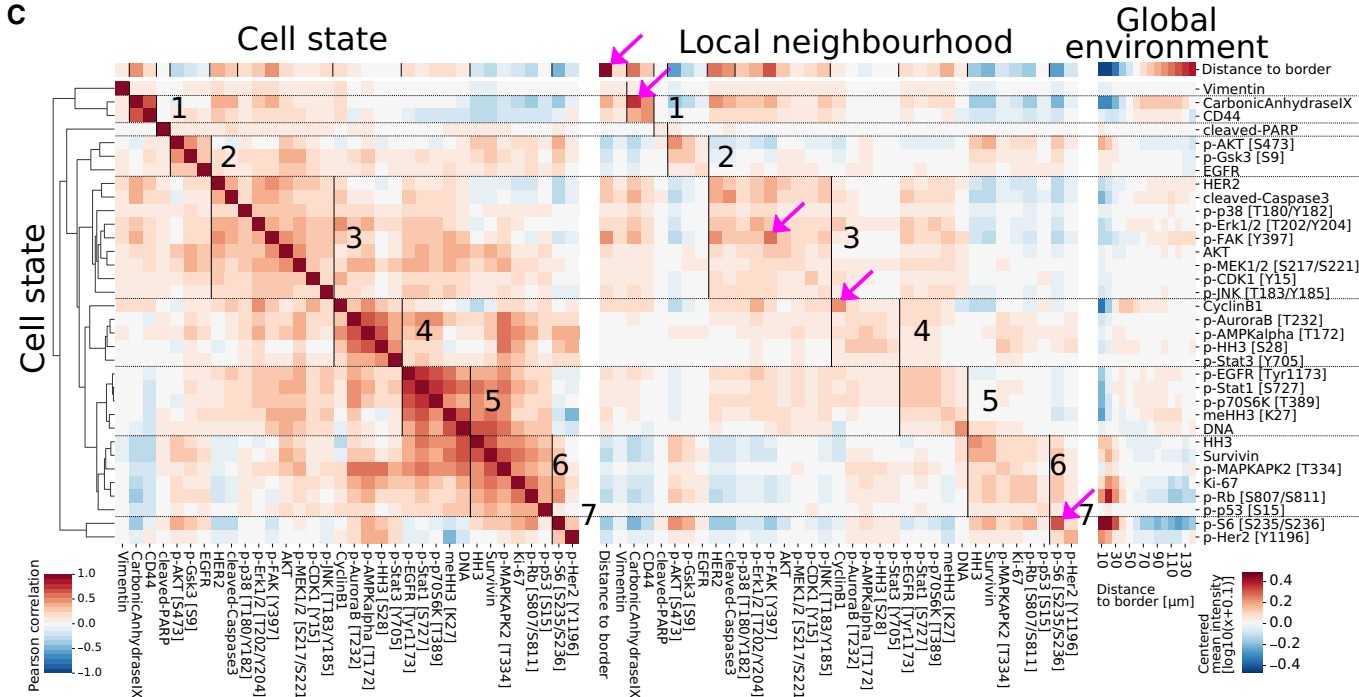

**Figure 2. Multiplexed imaging captures spatial organization of spheroids.**

A Example IMC images of a pooled spheroid plug (top row) and a HT-29 spheroid section (bottom row).

B Examples of image quantification showing log10-transformed average counts per cell section for the indicated markers.

C Correlation analysis of HT-29 spheroids (96 h growth). Left: Symmetrical Pearson's correlation matrix of markers within each cell. Clusters (indicated by horizontal lines and labeled with numbers) are based on hierarchical clustering of the intracellular marker correlation (distance cosine, metric average linkage). Middle: Correlation matrix of markers in all cells (rows) and average marker levels in neighboring cells (columns). Right: Median log10 intracellular marker levels as a function of the distance to the spheroid border. Values centered around 0. Pink arrows highlight strong spatial autocorrelations (Pearson's $r > 0.5$).

Hierarchical clustering of the intracellular marker correlation matrix identified seven clusters (Fig 2C, left). Clusters 2, 3, 5, 6, and 7 contain markers of activated growth signaling in the EGF and AKT/mTOR pathway, and cell-cycle markers. Mitotic markers are found in cluster 4. Cluster 1 consists of the classical hypoxia marker carbonic anhydrase 9 and the cell-to-cell adhesion marker CD44.

Vimentin and cleaved PARP, which were virtually absent in these spheres, did not cluster with other markers. Markers within the same cluster are consistently positive or negatively correlated with distance to border (Fig 2C, top row). For instance, all markers in cluster 3, containing EGF signaling and other markers, were positively correlated with distance to border, indicating these markers

seem to be co-expressed predominantly in the inside of spheroids. Conversely, all markers of cluster 6, containing cell-cycle markers such as Ki67 and p-RB, were negatively correlated with distance to border, indicating co-occurrence at the sphere border. These patterns suggest that the intracellular states captured by the clusters are linked to the spatial position within the sphere.

We next asked how these clusters mapped onto correlations between markers in neighboring cells. We correlated intracellular marker levels with the average marker levels of all cellular neighbors and ordered the resulting correlation heat maps according to the clustering derived from intracellular marker correlations (Fig 2C, middle). This ordering was in agreement with correlations between intracellular markers and average marker levels in neighboring cells, suggesting that intracellular marker correlations also capture correlations with the local neighborhood. Marker levels averaged over neighboring cells were even more strongly correlated with distance to spheroid border than were the intracellular levels, indicating that the local neighborhood is strongly dependent on the spatial position in the sphere.

We next focused on spatial autocorrelations (i.e., the correlations between an intracellular readout and the same readout in neighboring cells) (Fig 2C, middle, entries on diagonal). Low autocorrelation is indicative of markers being locally variable, while high autocorrelation suggests either that a marker occurs in cell patches or varies smoothly in the local cell neighborhoods. Most readouts had weak-to-medium spatial autocorrelation, but four had strong autocorrelations (pink arrows; Pearson's $r > 0.5$). The strongest autocorrelation was found for the distance-to-border readout, our surrogate measurement for the global environment; unsurprisingly, this was almost perfectly correlated with the average distance to border of neighboring cells. The other three strongly autocorrelated markers, p-S6, carbonic anhydrase, and p-FAK, were also all highly correlated with the distance-to-border measure (Pearson's $r$ with distance to border $> 0.5$); these gradients of expression were confirmed visually in spheroid sections (Fig 2A and B). Thus, spatial autocorrelation can capture effects of the global environment. However, low spatial autocorrelation of a marker does not necessarily imply a lack of influence by the global environment. For example, p-Rb, a marker of cells that have completed the G1/S transition, showed a strong distance-to-border effect (Fig 2C, right), yet only a moderate autocorrelation (Pearson's $r = 0.35$). This low local autocorrelation suggests that cells in a local neighborhood do not progress through a cell cycle in a synchronized manner even though our data overall show that the position of a cell in the global gradients determines its likelihood of being in a certain cell-cycle state.

Direct visualization of average marker levels as a function of distance to border confirmed that clusters defined by intracellular correlations show similar marker localization patterns (Fig 2C, right). This supports our hypothesis that intracellular marker correlations capture elements of the global environment (i.e., spatial position within the spheroid). We also observed a spatial segregation between markers of growth signaling, early cell cycle, and late cell cycle in all cell lines: AKT/mTOR signaling peaked in the outermost sphere layer, early cell-cycle markers (p-RB, Ki67) were located in the penultimate layers, and markers of the late cell cycle (cyclin B1) were generally located in the middle layers of the sphere (Fig EV4D). Thus, cellular states carry information about the spatial position of a cell within a sphere. Taken together, our analysis

indicates that intracellular markers are not only correlated within cells but that these states are also closely related to the cellular states of neighbors and the spatial location of cells in the global environment.

## Measurements of internal cell state, local environment, and global environment are interdependent

Given the strong and highly structured correlations observed, we asked to what degree marker levels are predictable by environment, local neighborhood, and cell state. We used linear modeling to predict the levels of each marker based on different predictive modules: the *global environment module* (a nonlinear function of the distance to border), the l*ocal neighborhood module* (the average marker levels of direct neighbors without autocorrelation), the *local autocorrelation module* (average marker levels of the predicted marker in immediate neighbors), and the *internal cell state module* (all other internal markers) (Fig 3A). In 56% of cases, the linear model including all modules explained more than 50% of the marker variability (Fig 3B). With the exception of few highly cell line-specific markers, total marker variability explained was usually similar for the different cell lines. In the best cases, the model explained about 85% of the total variation. The residual unexplained variance likely reflects a combination of technical variability in staining, detection, and quantification, the biological variability, and the inability of the linear model to capture nonlinear marker relationships. There was a clear relationship between average predictability and signal intensity for low-intensity markers (Fig EV5A), but not for markers expressed at medium- to high-intensity levels (higher than 1 average count per cell pixel). Thus, technical noise likely dominated the detection of the low-intensity markers.

Next, we investigated the explanatory power of the individual modules (Fig 3C–F). We expected that modules would not be independent in their explanatory power due to properties that result from the spatial tissue architecture: A cell and its neighbors, by virtue of their proximity, are subject to very similar global environmental cues. The global environment will thus similarly influence marker expression in a cell and its neighbors, leading to an indirect correlation between the two (Fig 3C). We therefore expected that measurements of the local neighborhood should also capture marker variability caused by the global environment. This was strongly supported by our data: The linear model based on the global environment module alone explained a median of 8.0% of variation. The local neighborhood module alone explained a median of 12.8% of variation. Adding the global environment module to a model containing the local neighborhood module only improved the predictive power by a factor of 1.12, an increase of only + 1.5% additional variability explained (Fig 3F top, Fig EV5C). This indicates that indeed the local neighborhood largely captures the global environment in the ability to explain marker variation.

By similar reasoning, if the expression of a marker in a given cell is strongly determined by the local and global environments, levels will be similar in neighboring cells (i.e., it is likely to be spatially autocorrelated). In this case, the local environment influences the expression of a given marker both in the cell of interest and in its neighbors (Fig 3D), and autocorrelation alone should explain a substantial fraction of marker variation caused by local and global

neighborhood effects. Supporting this hypothesis, local autocorrelation alone explained a median of 12% of marker variability in our data. The global environment and local neighborhood features together explained a median of 15% of marker variability. Adding these features to a model based on local autocorrelation improved the predictive power by 1.38-fold (+4.1%). This indicates that local

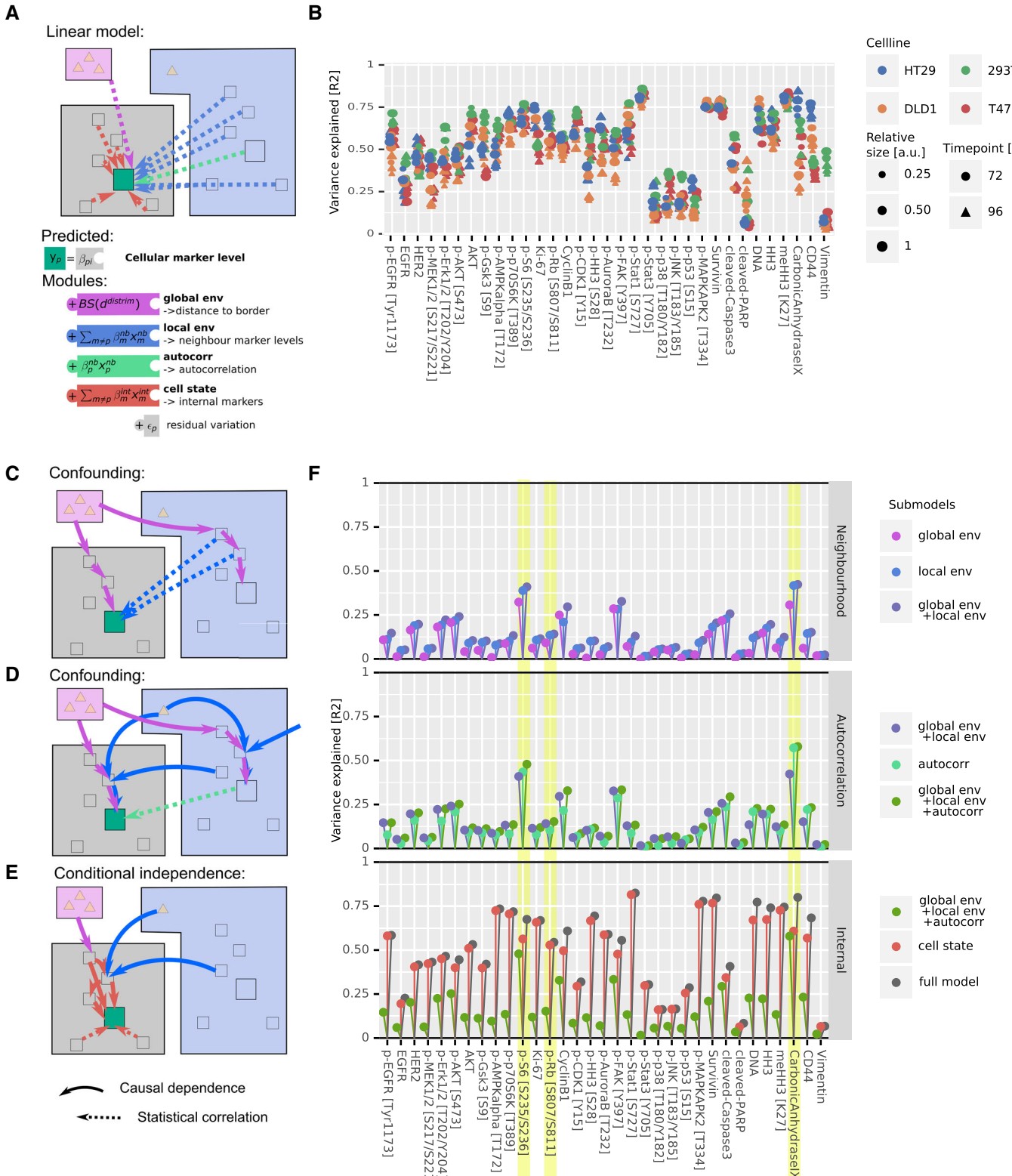

**Figure 3.**

**Figure 3.   Global environment, local neighborhood, and cell state are not independent predictors of single-cell marker levels in 3D spheroids.**

A   Marker levels predicted with a linear model using modules representing global environment (violet in schematic), local neighborhood (blue), autocorrelation (teal), and cell state (red). Squares represent protein marker states, and triangles represent nutrients or secreted growth factors.

B   Variance explained by the full model plotted for each marker, for all cell lines, and for all growth conditions.

C   The schematic depicts a confounding effect, through which a marker in a cell (green square) can be indirectly correlated with neighboring cell markers (dashed blue arrows) due to the global environment (violet arrows) affecting both cells and their neighbors.

D   Schematic depicting how confounding can cause a marker (green square) strongly dependent on the local and global environment to be statistically autocorrelated in neighboring cells (dashed teal arrow).

E   Schematic depicting how environmental influences on marker levels are transmitted via other intracellular proteins. Thus certain internal marker levels do capture environmental effects (red arrows).

F   Variance explained by the indicated modules for all markers in all cell lines and growth conditions. The data are visualized to illustrate the minimal added explanatory power of the local neighborhood over global environment (top), of autocorrelation over other spatial factors (middle), and of internal cell state markers over all environmental factors (bottom). p-S6, p-Rb, and carbonic anhydrase are highlighted examples (see also Fig EV5B).

Data information: For all schematics (C-E), bold arrows indicate a direct effect and dotted arrows indicate indirect statistical correlations.

autocorrelation alone captures around two thirds of the variability explained by spatial effects (Fig 3F middle, Fig EV5B).

Finally, since cells convert external stimuli into an intracellular response via a highly interconnected intracellular signaling network, we expected that environmental effects would not only influence the expression of markers determined directly by the environment, but also influence the expression of related internal markers (Fig 3E). Thus, a comprehensively measured internal cell state should capture much of the marker variability caused by the environment and neighborhood. This effect was indeed seen in our dataset: The internal cell state markers alone explained a median of 47% of variability, whereas all environmental terms together explained 17% of variability. Adding the environmental modules to the internal cell state module (to yield the full model) explained a median of only 1.05-fold more variability (+1.9%) than did the internal cell state module alone. Further, a model based solely on the internal cell state module captured more variability than a model with all neighborhood terms in 97% of cases (Fig 3F, bottom, Fig EV5C).

Analyses of three markers illustrate these patterns of increasing explanatory power as different modules are added to the model for HT-29 cells (Fig EV5B). Carbonic anhydrase IX (CA9) is a hypoxia marker, and its expression is known to depend on environmental conditions (Lal *et al*, 2001). Consistent with its role as a hypoxia marker, CA9 expression was observed in the sphere center (Fig 2). Although 31% of CA9 variation was explained by the global environment, the local neighborhood alone and spatial autocorrelation alone explained more variability (39% and 43%, respectively; Figs 3 F and EV5B). Adding the global environment module to a model containing these local readouts barely improved the predictive power (+0.1%). The internal state module alone predicted 61% of CA9 variation, whereas all environmental features together only predicted 58%. One might naively interpret these data on the basis of explanatory power to conclude that, since cell state alone explains more variability than all the spatial readouts, CA9 is largely dependent on internal cell state. Or, since spatial autocorrelation explains more variability than local neighborhood or global environment, one could conclude that autocorrelation is the most important spatial effect. However, we independently know that CA9 is environmentally determined (Lal *et al*, 2001). Thus, interpretations solely based on the explanatory power of features and that do not take into account their interdependence would miss the key biological dependence of this marker on the environment.

Similarly pS6, a growth marker, is dependent on the global environment since it is present at the highest levels in the outermost rim of the sphere consistent with its role in nutrient signaling (Fig 4C) (Manning & Toker, 2017). The explanatory powers of independent features suggest that pS6 depends more strongly on the local neighborhood than the global environment and even more strongly on the intracellular cell state (Fig 3F). However, accounting for the interdependencies between these factors, it becomes clear that the expression of pS6 is largely determined by the environment (Fig EV5B).

Finally, we found that also cell-cycle markers such as p-Rb are spatially segregated in the spheroids (Fig EV4D) and that 10% of the variability is explained by the global environment. This effect was largely captured by the local neighborhood and by internal cell state (Fig 3F). It is not surprising that a cell-cycle marker is predicted by internal cell state markers; however, treating the predictive factors as independent entirely masked the environmental contributions to marker variation. Had we not accounted for the interdependency between these factors, the spatial dependence of cell cycle in spheroids would have been missed (Fig EV5B).

In summary, a linear model based on measured global, local, and internal cell state features predicted a substantial fraction (an average of 50% and up to 85%) of single-cell marker variance in homogenous 3D spheroids. Our data strongly support our conceptual model-derived hypothesis that global environmental features, local environmental features, and intracellular features are interdependent in their ability to predict marker variation.

**Step-wise regression captures hierarchy of environmental marker dependencies**

The interdependencies we identified in the ability of different modules to predict marker variation appear to follow a hierarchy, with the explanatory power of the global environment captured by that of the local environment, which in turn is captured by that of intracellular features. We exploited this hierarchy to derive a concise visualization of the factors influencing marker variability. We reasoned that a biologically informative representation would indicate additional variability explained as each module is added step-wise to a regression model (Williams, 1978; Kruskal, 1987). The increasing order of explanatory power we observed, which also supports our conceptual model of cells interacting in tissue (Figs 1A and 3C–E), suggests that submodules should be added in the order

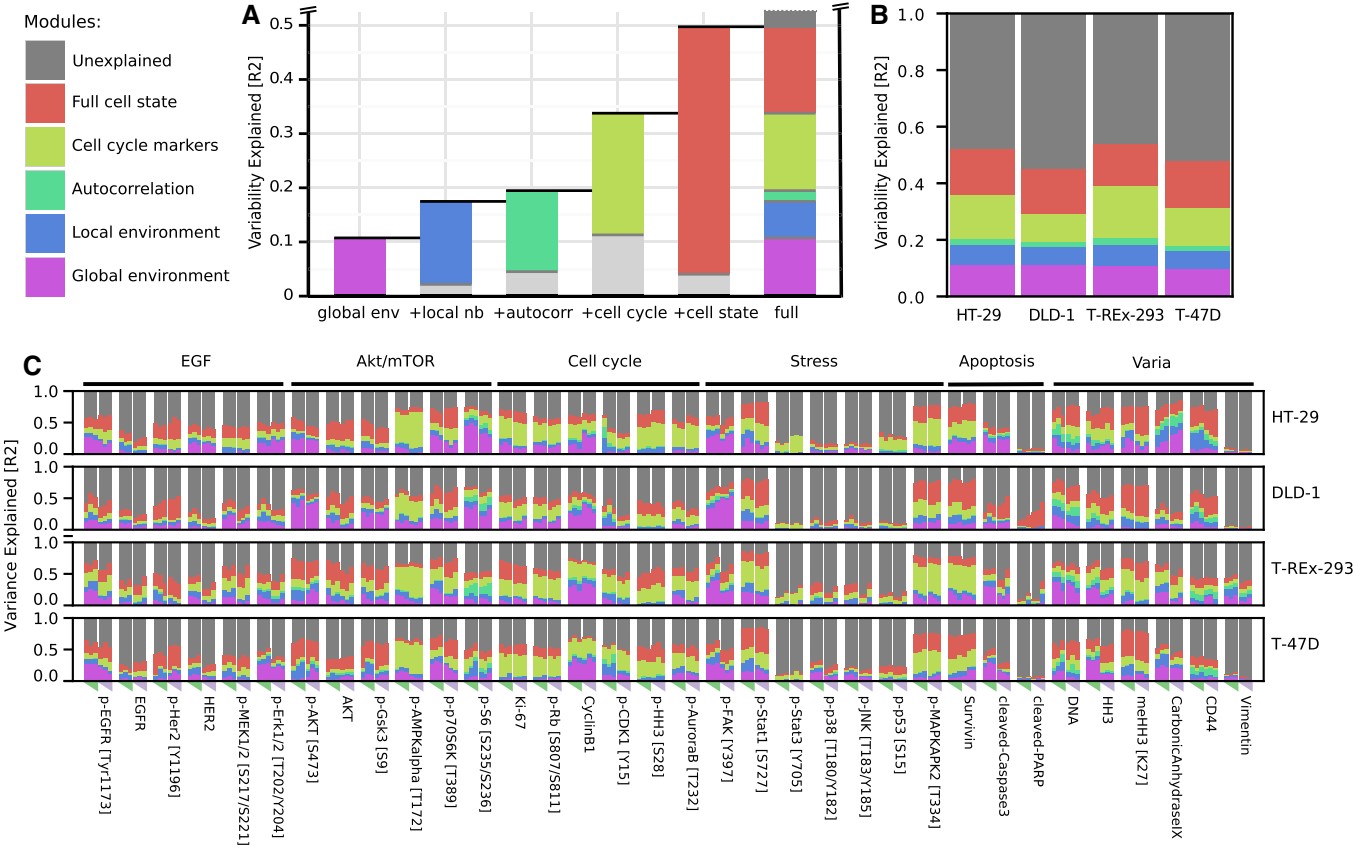

**Figure 4. Marker variance is hierarchically explained by cell-intrinsic and environmental factors.**

A Average marker variability over all markers explained by global environment, local neighborhood, autocorrelation, cell-cycle markers, and all intracellular markers. For each bar, colored portions indicate the variability explained by the particular module alone and the light gray portion indicates the additional variability explained when the previous module or modules are also included in the model. Dark gray indicates unexplained variance. Since the variability explained by each feature is not additive but roughly follows a hierarchy, the contributions to the full model are represented as a stacked bar plot.

B Contributions of the different modules to marker variance for each cell line, averaged over all markers and growth conditions.

C Contributions of the different modules to marker variance for each cell line and growth condition. Rows represent cell lines. Columns show marker abundances at a specific growth condition. Columns represent growth conditions varied by sphere sizes (triangle, 0.25/0.5/1.0x cells) and growth time (72 h green, 96 h pink).

of global environment, local neighborhood, autocorrelation, and internal cell state (Fig 4A). Exhaustive permutations indeed confirmed that this sequence (of all possible sequences of step-wise addition of these modules) optimally captures the contributions of all factors (Fig EV5E). Other sequences of module addition either mask the contributions of some modules or incorrectly exaggerate the contributions of others, as seen for p-S6 variation, for example (Fig EV5D).

We then used this representation to compare the marker variation explained by global environment, local neighborhood, and internal cell state for each marker across cell lines and growth conditions. Averaged over all cell lines, markers, and conditions, the linear model containing all modules explained 50% of variation, whereas 20% was explainable by all environmental factors (Fig 4A). Within the spatial effects (i.e., global environment, local neighborhood, and local autocorrelation), the global environment explained on average more than half (55%) of the variability. Averaging across all markers and growth conditions for each of the four cell lines showed similar dependencies (Fig 4B), suggesting that each of

these cell lines reacts similarly to internal and environmental influences when grown as 3D spheroids.

Our concise visualization based on the hierarchy of explanatory power also enabled fine-grained comparison of how each of the 34 markers depends on the global and local environments in four cell lines and under six growth conditions, allowing more than 4,000 comparisons (Fig 4C). We observed both general and cell line-specific effects. We note that, across the dataset, the average standard deviation of the explained variability was less than 0.04 for all models (overall average 0.036, iqr. 0.018–0.047) across five spheroid replicates for each of the 24 growth conditions.

Since the cell cycle is a major source of cell-to-cell variability (Gut *et al*, 2015; Buettner *et al*, 2015; Rapsomaniki *et al*, 2018), we further classified the internal markers into cell-cycle and non-cell-cycle markers (Dataset EV1). For a given cell-cycle marker, an average of around 50% of variability was explained by the full model (Fig 4C). Cell-cycle markers and environment together captured 75% of this variation. An exception across all cell lines was p-HH3, a mitotic marker, for which environment and cell-cycle markers

explained only 58% of all explainable variation. This suggests that mitosis is strongly linked to the cellular state as a whole and not only to cell-cycle markers. We also observed cell line-specific effects. For instance, Ki67 variability was strongly linked to non-cell-cycle intracellular markers specifically in T-REx-293 cells. Early cell-cycle markers, p-RB and Ki67, showed little dependence on the global environment in T-47D cells (approximately 2% variability explained), but the global environment explained 7-12% of the variability observed in HT29, DLD-1, and T-REx-293 cells. Of all the cell-cycle markers, cyclin B1 levels were the most dependent on the environment, with an average of more than 20% of variability explainable by global environmental gradients in all cell lines (Fig 4C).

The AKT/mTOR pathway is involved in growth and nutrient signaling (Manning & Toker, 2017). We found that the levels of multiple markers of this pathway are explained by environment and local neighborhood features. A downstream readout of this pathway, p-S6, was strongly dependent on environmental factors in all cell lines. Other upstream markers, such as p-AKT and p-GSK3beta, showed cell line-specific effects: Environmental factors had higher explanatory power for these markers in DLD-1 cells than in other cell lines. Finally, the levels of p-AMPK, reported to be a nutrient sensor (Mihaylova & Shaw, 2011), were strongly explained by the cell cycle but only slightly by environmental factors. This is also reflected in the correlation maps, which showed that p-AMPK expression was correlated with that of mitosis markers (Figs 2C and EV4A–C), consistent with the reported association of this marker with the mitotic spindle (Vazquez-Martin et al, 2009a, 2009b).

In summary, our analysis of multiplex imaging data in homogenous 3D tissue models allowed a detailed deconvolution of the factors affecting marker variation. Internal cell state, local neighborhood, and global environmental factors are interdependent and follow a hierarchical order of their explanatory power for marker variation. The variability explained by different factors was on average similar across cell lines. There were, however, impacts of cell line and growth conditions on expression levels of certain markers. Our data allow granular identification of these cell line-specific and growth condition-specific patterns in marker dependencies.

## Signaling deregulation affects cells and their neighbors in spheroids

Our experiments showed that, after correcting for global effects, on average 6% of marker variability was predicted by neighboring cell markers. To explore whether these correlations reflect spatial coordination due to active communication between cells, other biological effects, or technical artifacts, we developed an overexpression system to induce changes in individual cells and investigate the effects on neighboring cells. We hypothesized that in the overexpression context, active cell communication should lead to systematic changes in neighboring cell states. We used a previously described library of 32 pro-cancer signaling protein constructs involved in 17 pathways and containing many common cancer driver mutations (Martz et al, 2014), supplemented with ten growth factor receptors, nine ligands, and four negative controls (Dataset EV3). Inducible expression vectors for each GFP-tagged protein were individually transiently transfected into separate wells of T-REx-293 cells. Overexpression was induced during 24 h after spheroid formation (Fig 5A and B). Under the conditions used, overexpression usually occurred

in a subset of cells in a spheroid (Fig EV6A). We combined GFP detection using two independent antibodies to identify cells that overexpressed a particular protein (overexpressors), the direct neighbors of overexpressors that did not themselves overexpress the protein (neighbors), and non-overexpressing cells that were not neighbors of an overexpressing cell (bystanders) (Fig 5C). Further, we assigned weakly GFP-positive cells that were localized next to strongly overexpressing cells as ambiguous, since discriminating weak overexpression from spurious positivity due to spatial proximity was not possible. In total, we assessed six replicate spheres for each construct and 30 mock-transfected spheres as technical negative controls. We analyzed more than 500,000 cells from 1,968 spheroid sections from 278 spheroids (Dataset EV2), corresponding to an average of 7 random sections per sphere.

For each of the overexpression constructs, we tested whether overexpressor, neighbor, or bystander cells were significantly different in their marker expression from cells of mock-transfected spheres (linear mixed-effects model, $P < 0.01$, $q < 0.1$, fc > 20%). Whereas intracellular effects should be largely cell-autonomous, we expected that effects on direct neighbors should be dominated by a combination of juxtacrine and paracrine effects. Further, we assumed that bystanders are mainly affected by longer-range paracrine effects of cells in the measured plane and in the planes above and below the evaluated cell, though juxtacrine effects of off-plane cells could also plausibly contribute to bystander effects.

First, we examined cell-autonomous effects of overexpression. Compared with the mock-transfected control spheres, we observed a stress response in overexpressors for most constructs (p-p38: 86%, p-SAPK/JNK: 64%, Figs 5D and EV6B), including three of the four negative control constructs. A nonspecific stress response to overexpression was not unexpected (Moriya, 2015) and thus was not reported as an overexpression-specific effect or included in reported statistics except when explicitly mentioned. We observed that the overexpression of 23 of 32 intracellular signaling proteins, seven of nine ligands, and ten of ten receptors but none of the four negative controls significantly affected more than one intracellular marker (Figs 5D and EV6B). This indicates that the overexpression of most of the constructs perturbed the intracellular state.

The overexpression effects were often consistent with known functions of the overexpressed protein and usually involved multiple markers in the relevant pathway (Figs 5D and EV6B). For example, EGFR overexpression increased total EGFR and p-HER2 as expected (Fig EV6B) (Alroy & Yarden, 1997). FGF receptor overexpression strongly activated its downstream target p-ERK1/2 as well as p-EGFR and p-HER2 (Fig EV6B) as previously reported (Hinsby et al, 2003). TGF-beta and TGF-beta receptor 2 overexpression both reduced Ki67 and p-RB levels significantly (Fig 5D) as reported (Massagué, 2012). Reassuringly, in the three cases where an antibody in our panel detected the overexpressed protein, we detected significantly higher levels of the overexpressed proteins in cells transfected with the particular expression construct than in mock-transfected control cells. Where a phosphorylation site in the overexpressed protein was monitored, we observed an increase upon overexpression in three of four cases. The exception was EGFR overexpression, which increased total EGFR and phosphorylation of its interaction partner HER2 but surprisingly did not increase levels of p-EGFR. Overall, these data show that our approach detects biologically expected intracellular responses to overexpression.

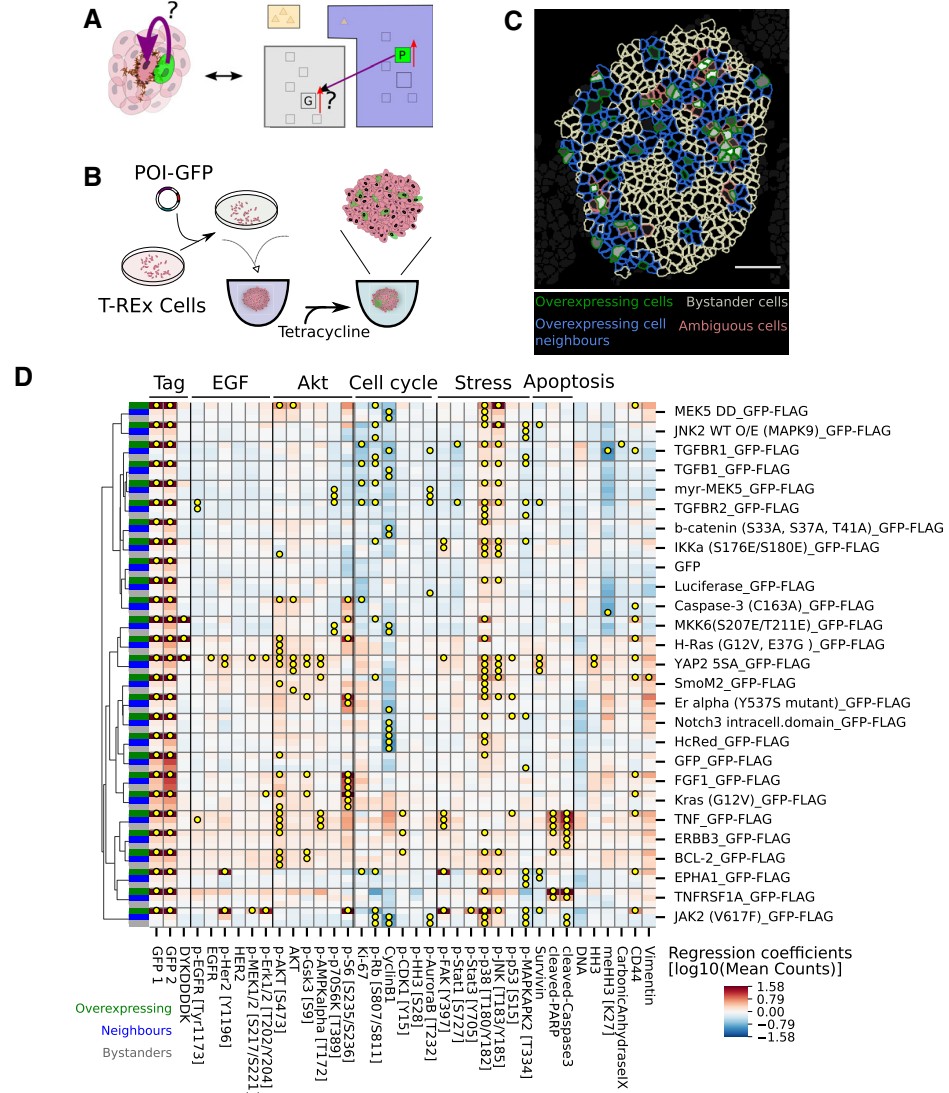

**Figure 5. Systematic overexpression reveals spatial effects of signaling deregulation.**

A   Left: Depiction of a spheroid with a cell overexpressing a construct of interest (green) that has an effect on a neighboring cell. Protein overexpression could have intracellular and neighborhood effects. Right: Illustration of an overexpression situation and the question of whether the overexpression of protein P (green) alters the expression of marker G in a neighboring cell.

B   A schematic of the overexpression system used in this study. Inducible transient transfection leads to GFP-tagged protein overexpression in a fraction of cells in spheroids.

C   A representative image of a spheroid is shown illustrating the identification of overexpressing cells (green), their neighbors (blue), and bystander cells (white). In cells classified as ambiguous (pink), we could not distinguish between overexpression in the cell itself and signal spillover from overexpressing neighbors. White scale bar indicates 50 μm.

D   Matrix of overexpression estimated effects of constructs (rows) on markers (columns) classified as intracellular ( green ), neighborhood ( blue ), and bystander (gray) in column at the far left. Yellow dots indicate strong, significant effects ($P < 0.01$, $q < 0.1$, fold change > 20%, neighbor/bystander effects: >0.1x internal effects, test: t-statistics for linear mixed-effects model coefficients using Satterthwaite's method for denominator degrees of freedom). All controls and constructs with neighborhood or bystander effects on more than one marker are shown here (see Fig EV6B for data for all constructs).

We next examined the effects of overexpression on neighboring cells, making use of the GFP intensity in neighbors as an additional criterion to account for spatial signal bleed-over. Of the 55 constructs tested, nine caused changes in more than one marker in neighboring cells and ten caused multiple changes in bystander cells (Fig 5D). Notably, all of these also caused specific and significant changes in internal cell state as well. Of the constructs that caused changes in the expression of at least two markers in neighboring cells, one was a ligand (of eight ligands tested), five were intracellular signaling proteins, and four were receptors (of 10 receptors tested) (Fig 5D). The effects on neighbors (marker fold change, iqr [1.2–1.5], max 2.4) and bystanders (iqr [1.2–1.4], max 2.0) were usually weaker than internal effects of overexpression (iqr 1.3–5.1, max 42.8).

We found that the overexpression of constitutively active YAP 5SA had the most profound effect on the measured marker panel; it affected 13 markers intracellularly and 7 markers in neighboring cells. The intracellular effects indicated an activated MAPK pathway (as indicated by increased EGFR, p-HER2, p-ERK, and p-MEK expression) and activated mTOR/AKT signaling (increased p-AKT, AKT, pGSK3Beta, and p-AMPKalpha expression). Both these pathways are thought to be upstream of YAP, thus indicating an intracellular positive feedback loop (Basu *et al*, 2003; He *et al*, 2015). The overexpression of constitutively active YAP also affected AKT signaling and p-HER2 in neighbors indicating an intercellular effect. We speculate that this combination of intra- and intercellular signaling could be mechanistically explained by the excretion of a ligand that elicits both autocrine and paracrine effects, consistent with the autocrine loops reported for this pathway (He *et al*, 2015; Rizvi *et al*, 2016). However, the overexpression of two ligands that are transcriptional targets of YAP, AREG and FGF1, and that have been suggested to be involved in these autocrine signaling loops, did not elicit the same effects as YAP 5SA overexpression.

We found that, of the overexpressed ligands, only TNF had strong effects on more than one marker in neighboring cells. TNF overexpression induced apoptosis throughout the sphere, in both neighbor and bystander cells (Figs 4C and EV6). In summary, these perturbation experiments demonstrated how coupling multiplexed imaging to 3D tissue culture can be used to study non-cell-autonomous effects of signaling deregulation, providing insight into the factors determining spatial relationships between markers and thus into the mechanisms underlying cellular organization.

# Discussion

### Cell state and environmental measures influence cellular phenotypes in an interdependent manner

We coupled a 3D spheroid tissue model system with highly multiplexed imaging to characterize the influence of global and local cellular environment on cellular phenotypes. We observed that measures of local and global environments and internal cell state are not independent in their abilities to predict marker variation. Rather, there were strong nonadditive interdependencies among these factors. Specifically, and consistent with the spatial architecture of spheroids, measurements of the local neighborhood of a cell captured marker variability explained by the global environment. Spatial autocorrelation alone explained much of the marker variation captured by local and global environmental effects. Finally, intracellular state markers (including cell-cycle markers) recapitulated much of the explanatory power of all environmental effects combined.

Such interdependencies must be taken into account in studies aiming to deconvolve the contributions of environmental factors to phenotypic variability. For example, although a comprehensive intracellular marker measurement predicts the behavior of an environmentally sensitive marker even without an environmental measure, this does not mean that such environmental effects do not exist. In fact, environmental effects could be the causal reason for the behavior of the marker, reflected in the fact that intracellular markers are accurate surrogates for environmental conditions in nonspatial cytometry analyses (Moon *et al*, 2007). Examples are

hypoxia markers as surrogates for cell position in an oxygen gradient and phosphorylated receptor levels as surrogates for ligand binding. We showed that environmental factors that affect marker expression (for instance, of the hypoxia marker CA9) are missed if the interdependence between explanatory factors is not taken into account.

Interdependencies in spatial measurements have been acknowledged in *E. coli* (van Vliet *et al*, 2018). However, a recent approach developed for multiplexed data analysis, spatial variance component analysis, assumes that contributions of spatial proximity (environment), neighborhood levels (cell-to-cell interactions), and cell state (intrinsic) are independently additive (Arnol *et al*, 2019). This assumption may bias results. To account for interdependencies in our own dataset, we used a step-wise regression approach, in which predictors were added in increasing order of explanatory power. We showed that this regression approach was able to quantify how phenotypic markers depend on the influence of the global environment, local neighborhood, autocorrelation, and internal markers. We confirmed several of the identified patterns by visual inspection of images. Our simple model system allowed us to compare the interdependent effects of environmental and cell-autonomous factors on cellular phenotype in different cell lines and growth conditions and to quantify marker-specific differences in spheroid organization.

Our antibody panel was chosen to examine markers expected to reflect heterogenous growth phenotypes and signaling in homogenous spheroids. The chosen marker panel will to some extent affect the model outcome; however, we observed that 50% up to 85% of marker variability was explained using our complete linear model, which included global environment, local neighborhood, local autocorrelation, and internal cell state modules. For low-abundance markers (< 1 average count per cell pixel), technical detection noise likely dominated marker variability (Fig EV5A). This was not the case for markers expressed at higher levels, however. An additional technical source of variability may result from our reliance on 6-μm-thick slices of cells, measured at a lateral resolution of 1x1 μm. At this resolution, pixels may belong to more than one cell, and segmentation is unlikely to be perfect, which introduces technical variability. Further, our readouts do not represent full cells but random, 6-μm-thick slices through cells, which could introduce technical sampling variability, in particular when markers are not uniformly distributed across the cell. Finally, our analysis assumed linear marker relationships, which may explain the lack of fit of our model to some extent.

### Challenges in adapting the analysis to complex tissues

A future challenge will be to apply similar approaches to heterocellular tissues, which are more representative models of biological systems than the spheroids analyzed here. Though the specific factors affecting cellular phenotypes in a particular tumor context will vary, it is likely that the interdependencies we have identified between intracellular and local and global environmental factors will remain valid and could inform spatial analyses in more heterogenous tissues as well. Such tissues are likely to be highly structured with different cell types confined to specific locations, resulting in strong cell-type co-occurrence patterns. Applying methods that quantify relationships between cells and their neighborhood, agnostic of cell types, will likely capture cell-type co-occurrences as neighborhood effects (Arnol *et al*, 2019). Although meaningful, co-occurrence of cell types does not provide the full

picture of how the cellular phenotype is influenced by the environment or neighborhood. Lineaging approaches could help mitigate the confounding effects of co-occurring cell types and provide a ground truth for phenotypically comparable cells.

Identification of biologically relevant spatial gradients in heterocellular tissues will be much more challenging than in our symmetrical spheroid model, which allowed estimating these gradients based on the known location of the source (i.e., nutrients in the medium) and of the sink (i.e., cells). Gradient characterization will be important, as we illustrated here for spheroids, where a readout for such gradients is key to understanding the cause of observed spatial variability and correlations. Although it may be theoretically possible to estimate the number of relevant biological gradients in complex tissue based only on phenotypic information (Adler *et al*, 2019), capturing quantitative information on these gradients will require a stereotypical tissue structure and biological domain knowledge. Identifying gradients in tissues and using them as biologically relevant coordinate systems will aid in identification of causes of phenotypic variability and will enable comparisons across tissue samples.

In summary, although modeling influences on phenotypic plasticity in tumor tissue will be challenging due to complexities of cell type and lineage, co-localization due to structured tissues, and unknown global environmental gradients, we expect that insights gained from the simplified spheroid systems will inform accurate spatial analyses of phenotypic variation in more complex systems.

### The influence of extreme cell states on neighboring cells

We used a chimeric overexpression system to systematically assess the effects of deregulated signaling on cellular neighborhoods in the spheroids formed by T-REx-293 cells. Of the 55 constructs overexpressed, 73% induced intracellular changes in multiple markers, and around 20% caused non-cell-autonomous effects on neighboring and bystander cells. This indicates that the chronic overexpression of signaling proteins alters not only the intracellular state of the cell overexpressing the signaling protein but also, at least in some instances, cell states of neighbors.

It is likely that our analysis missed some effects: Although our marker panel covers multiple signaling pathways and cellular processes, our previous studies have shown that overexpression can alter signaling transiently, without an effect on steady-state marker levels at the time of measurement (Lun *et al*, 2017, 2019). Focusing on steady-state levels in our analysis meant that we missed such dynamic effects. Further, misfolding and mislocalization of tagged, overexpressed constructs can lead to nonphysiological effects, including the stereotypic intracellular stress responses evident in

our data. We observed intracellular responses consistent with known biological functions of many overexpressed proteins, but cannot rule out that some of the constructs were misfolded or mislocalized in some way.

The use of linear mixed-effects models allowed us to take into account dependencies due to the experimental design and due to global environmental effects, thus increasing the reliability of the results. However, we did assume normality, heteroscedasticity, and spatial independence of residuals. These assumptions are violated to various degrees, potentially leading to false positives and false negatives. Despite these theoretical reservations, the reliability of our results is supported by the finding that the expression of negative control constructs (two different GFP constructs, HcRed, and luciferase) did not significantly change intracellularly or in neighbors in more than one marker apart from the stereotypic stress responses, whereas 80% of overexpressed proteins did.

There are multiple potential extensions of these methods to analyze this spatial overexpression dataset. Apart from statistically better modeling of the spatial dependencies, these data would also be suitable to investigate more complex phenomena such as reciprocal signaling, a phenomenon that has been previously described in co-cultures, in which cells react differently to overexpression depending on their neighborhood (Tape *et al*, 2016).

In conclusion, we developed a novel tissue barcoding workflow for simultaneous processing of up to 240 microtissues and used this setup to generate a large multiplexed imaging dataset of homogeneous 3D spheroids with single-cell resolution. Our dataset will be a useful resource for the further development of algorithmic approaches describing spatial variability in cellular phenotypes. We have assessed how cell state and local and global environment affect cellular phenotype and report hierarchical interdependencies of these factors in their ability to explain marker expression. Our approach is broadly applicable and with appropriate methodological modifications will enable the robust characterization of more complex tissues, from co-cultures to heterocellular organoids and small embryos. Importantly, the interdependence of local and environmental factors that we demonstrated in the simple spheroid system must be taken into account in systems-level spatial studies of heterogenous tissues. We also demonstrated that our approach is compatible with perturbation studies and identified cell-autonomous and neighborhood effects of overexpressed cancer-related signaling proteins. We envision that this approach could be used to systematically study the impact of perturbations on the organization of simple and complex microtissues. This strategy could, for example, provide insight into how drug treatment alters the interplay of cell types in healthy and diseased tissue.

# Materials and Methods

### Reagents and Tools table

| Reagent/Resource | Reference or source | Identifier or catalog number |
| --- | --- | --- |
| **Experimental models** | | |
| T-REx-293 | Source: Invitrogen | R71007, STR: 100% match with HEK293.2sus (ATCC® CRL-1573.3™) |

**Reagents and Tools table** (continued)

| Reagent/Resource | Reference or source | Identifier or catalog number |
|---|---|---|
| Flp-In T-Rex DLD-1 | Source: Donation Stephen Taylor lab, University of Manchester | R71007, STR: 100% match with DLD-1 (ATCC® CCL-221™) |
| HT-29 | Source: NCI-Frederick Cancer DCTD Tumor/Cell Line Repository | STR: 100% match with HT-29 (ATCC® HTB-38™) |
| T-47D | Source: ATCC | STR: 100% match with T-47D (ATCC® HTB-133™) |
| **Recombinant DNA** | | |
| Constructs are listed in Dataset EV2 | | |
| **Antibodies** | | |
| Antibodies are listed in Dataset EV1 | | |
| **Oligonucleotides and sequence-based reagents** | | |
| pDEST pcDNA5 FRT TO-eGFP | Source: Anne-Claude Gingras (Lunenfeld-Tanenbaum Research Institute, Toronto, Canada, Reference: Couzens *et al*, 2013 | |
| pDEST 3' Triple Flag pcDNA5 FRT TO | Source: Anne-Claude Gingras (Lunenfeld-Tanenbaum Research Institute, Toronto, Canada, Reference: Couzens *et al*, 2013 | |
| **Chemicals, enzymes and other reagents** | | |
| High-glucose DMEM | Sigma | D5671 |
| RPMI-1640 | Sigma | R0883 |
| Penicillin-Streptomycin-Glutamine | Gibco | #10378016 |
| Insulin solution human | Sigma | I9278 |
| TrypLE™ Express Enzyme | Gibco | #12605010 |
| 0.2 μm vacuum filter | Nalgene, Thermo | #564-0020 |
| MycoAlert PLUS Mycoplasma Detection Kit | Lonza | LT07-703 |
| Maxpar® X8 Multimetal Labeling Kit | Fluidigm | #201300 |
| Antibody Stabilizer PBS | Candor | #131 050 |
| Fetal Bovine Serum (FBS) | Gibco | Heat Inactivated FBS, #10500 |
| tetracycline-free FBS | Biowest | S182T |
| 60 well BC scheme | adapted from Zunder *et al*, 2015, Bodenmiller *et al*, 2012 | |
| 126 well BC scheme | adapted from Lun *et al*, 2019, Zunder *et al*, 2015, Bodenmiller *et al*, 2012 | |
| PBS | Gibco | DPBS (1x), 14190-94 |
| 16% PFA | Electron Microscopy Sciences | #15710 |
| Breathe Easier | Diversified Biotech | BERM-2000 |
| Monoisotopic Cisplatin Pt194 | Fluidigm | #201194 Cell-ID Cisplatin-194Pt |
| Monoisotopic Cisplatin Pt198 | Fluidigm | #201198 Cell-ID Cisplatin-198Pt |
| Bovine Serum Albumin (BSA) | Sigma | heat shock fraction, pH 7, ≥98%, A7906 |
| 200ul wide bore tips | Corning Axygen | FX-255-WB-R |
| Gelatine | Dr Oetker | Gold Extra Sheets, B000FRSRJE |
| 10% Sodium Azide | Merck | # 26628-22-8 |
| UltraPure Agarose | Invitrogen | # 16500100 |
| 0.1M PB pH 7.4 | adapted from Recipe PB (0.1 M phosphate buffer pH 7.2), Cold Spring Harb Protoc 2010 | https://doi.org/10.1101/pdb.rec12291 |
| Sucrose | Sigma | BioXtra, S7903 |
| Tryphan Blue 0.4% | Invitrogen | T10282 |
| Tissue-Tek® O.C.T.™ Compound | Sakura | #4583 |
| 2-Methylbutane | Sigma-Aldrich | ReagentPlus®, ≥99%, M32631 |

Reagents and Tools table   (continued)

| Reagent/Resource | Reference or source | Identifier or catalog number |
|---|---|---|
| SuperFrost Plus™ Adhesion slides | Thermo Scientific | Thermo Scientific™ J1800AMNZ |
| 96-well Ultra-Low Attachment Spheroid Microplate | Corning | #4515 |
| Trizma® base | Sigma-Aldrich | #93350 |
| Sodium chloride | Sigma-Aldrich | ReagentPlus®, ≥99%, S9625 |
| Dako Pen | Agilent | S200230-2 |
| Tween-20 | Sigma-Aldrich | P9416 |
| Cell-ID™ Intercalator-Ir | Fluidigm | # 201192A |
| Hoechst 33342 | Invitrogen | H3570 |
| Telox 2 | donation from Nitz lab, Edgar *et al*, 2016 | |
| Jet Prime | Polyplus | #114 |
| TrypLE Select Enzyme | Gibco | # A1217701 |
| Tetracycline Hydrochloride | Sigma | T7660 |
| **Software** | | |
| CATALYST | https://doi.org/doi:10.18129/B9.bioc.CATALYST | v1.10 |
| imctools | https://doi.org/doi:10.5281/zenodo.3973063 | v1.0.7 |
| ImcPluginsCP | https://10.5281/zenodo.4057958 | v1.3 |
| CellProfiler | https://doi.org/10.1371/journal.pbio.2005970 | v3.1.8 |
| Ilastik | https://doi.org/10.1038/s41592-019-0582-9 | v1.3.2b3 |
| TrakEM2 | https://doi.org/10.1371/journal.pone.0038011 | v1.0i |
| scanpy | https://doi.org/10.1186/s13059-017-1382-0 | v1.6.0 |
| anndata | https://doi.org/10.1186/s13059-017-1382-0 | v0.7.4 |
| SciPy | https://doi.org/10.1038/s41592-019-0686-2 | v1.5.2 |
| Statsmodels | Seabold & Perktold, 2010 | v0.12.0 |
| Matplotlib | https://zenodo.org/record/3264781#.X2hR3_HgrmE | v3.3.2 |
| numpy | https://doi.org/10.1038/s41586-020-2649-2 | v1.19.1 |
| Snakemake | https://doi.org/10.1093/bioinformatics/bts480 | v5.18 |
| spherpro | This study, https://github.com/BodenmillerGroup/spherpro | v0.9 |
| Singularity | https://doi.org/10.5281/zenodo.3234175 | v3.2.1 |
| Python | www.python.org | v3.7.7 |
| **Other** | | |
| Biomek FX | Beckmann Coulter | |
| Hyperion Imaging Mass Cytometer | Fluidigm | |
| Countess | Invitrogen | |
| ImageXpress Micro XL Widefield High Content Imaging microscope | Molecular Devices | 4x objective, NA 0.20 |
| Axioscan Slide Scanner Z1 | Zeiss | |

## Methods and Protocols

### Cell lines

T-REx-293 cells (Invitrogen) and DLD-1 cells (Flp-In T-Rex DLD-1, a kind gift from the Stephen Taylor Lab, University of Manchester) were grown in high-glucose DMEM (D5671, Sigma). HT-29 (ATCC HTB-38) and T-47D (ATCC HTB-133) cells were grown in RPMI-1640 medium (R0883, Sigma). The media were supplemented with 100 U/ml penicillin, 100 mg/ml streptomycin, and 2 mM L-glutamine (Gibco, Invitrogen) and 10% fetal bovine serum (Gibco for T-47D, HT-29, and DLD-1 cultures, Biowest for T-REx-293). For T-47D cells, 0.2 U/ml human insulin was added. All media were filtered through a 0.2-μm membrane (Nalgene, Thermo). 1x TrypLE Express (Life Technologies) was used for cell passaging and harvesting. Cells were tested for mycoplasma with a MycoAlert PLUS Mycoplasma Detection Kit (Lonza). All cell line identities were verified using STR profiling (Microsynth).

### Antibody conjugation

Isotope-labeled antibodies were prepared using the manufacturer's standard protocol using the MaxPAR Antibody Conjugation Kit (Fluidigm). Conjugated antibody yield was determined based on absorbance at 280 nm. For long-term storage, antibodies were stored at 4°C in PBS Antibody Stabilization Solution (Candor).

### Spheroid cultivation

Preparation:

1   Filter all media using a 0.2-μm vacuum filter (Nalgene, Thermo) to avoid particles.
2   Prepare calculations for final dilutions and prepare a dilution series.

Seeding:

1   Cultivate cells in 2D culture until ca 80% confluency.
2   Wash cells with 37°C PBS (Gibco).
3   Add TrypLE™ Express Enzyme (Gibco), incubate at 37°C until cells detach.
4   Quench with warm medium.
5   Take two 10 μl aliquots to count.
6   Spin cells down at 250 g for 4 min.
7   Meanwhile:
-   Count the cells in the aliquots using a cell counter (e.g., Countess (Invitrogen)).
-   Calculate the required dilutions.
8   Remove supernatant of cells.
9   Resuspend cells in warm media.
10  Optional: count again.
11  Dilute cells to final seeding concentration using a dilution series.
12  Seed 100ul of cell suspension to each well of the 96-well Ultra-Low Attachment Spheroid Microplate (Corning).
13  Spin plate 4 min at 250 g.
14  Optional: image plate using an automated bright-field microscope to verify seeded cell number.
15  Seal the plates using an breathable membrane (Breathe Easier, Diversified Biotech).
16  Incubate at 37°C and 5% $CO_2$.
17  Optional: Image plates using an ImageXpress Micro XL Widefield High Content Imaging Microscope (Molecular Devices, 4× objective, NA 0.20) each day to monitor growth.

### Spheroid harvesting

#### Bright-field imaging

Bright-field imaging of intact spheres was performed using an ImageXpress Micro XL Widefield High Content Imaging Microscope (Molecular Devices, 4× objective, NA 0.20) at multiple z-planes. Spheres were imaged 2 h before PFA fixation and after PBS washing the next morning. Plates were acquired twice, rotating the plate by 180° between data acquisition to avoid imaging artifacts.

#### PFA fixation

Optional: Telox 2 hypoxia assay

1   Prepare 200 μM Telox 2 in 2% DMSO (Edgar *et al*, 2016).
2   Add 5 μl of solution per well to grown spheroids.
3   Incubate for 4h in the incubator and fix using PFA (see below).

All pipetting steps were implemented with a Biomek FX Robot (Beckmann Coulter).

1   Fix spheres by adding 30 μl of 16% PFA (Electron Microscopy Sciences) per well.
2   Incubate shaking at 200 RPM for 5 min.
3   Store overnight at 4°C.
4   Optional: image plates using bright-field imaging.
5   Wash plate four times with 150 μl of 1× PBS using a Biomek Fx Robot.
6   Optional: image plates using bright-field imaging.

### Barcoding and pooling

#### Barcoding schemes

1   60-well barcoding scheme used for 4 cell line dataset: prepared according to (Zunder *et al*, 2015): 8 choose 4 barcoding scheme with following metals and stock concentrations: 102Pd (10 μM), 104Pd (15 μM), 105Pd (20 μM),106Pd (20 μM), 108Pd (20 μM), 110Pd (15 μM), 113In (20 μM), and 115In (20 μM) in DMSO (Sigma).
2   126-well barcoding scheme used for overexpression dataset: prepared according to (Zunder *et al*, 2015): 9 choose 4 barcoding scheme with following metals and stock concentrations: 89Y (10 μM), 103Rh (200 mM), 105Pd (10 μM), 106Pd (10 μM), 108Pd (10 μM), 110Pd (10 μM), 113In (20 μM), 115In (10 μM), and 209Bi (2 μM) in DMSO.

To extend the barcoding capacity, spheres from multiple plates are collected and either barcoded with different monoisotopic cisplatin (Pt 198, Pt194, Fluidigm).

#### Barcoding

All pipetting steps were implemented with a Biomek Fx Robot (Beckmann Coulter).

1   Remove PBS from washing by sucking all liquid at a height of ca 2 mm from well bottom from the middle of the well using a gentle flow rate (estimated residual volume ca 30 μl).
2   Pre-dilute 4 ul barcoding solution with 65 μl of PBS and add to each well.
3   Incubate plates for 1 h shaking at 200 RPM.
4   Wash plates four times with 150 μl of 1x Cell Staining Medium (CSM, PBS (pH 7.4, Gibco) 0.5% bovine serum albumin (Sigma)).

#### Pooling

1   Incubate collection tubes with CSM for 10 min -> use 1 collection tube per cisplatin barcode.
2   Remove supernatant from collection tube and pool spheres from 96-well plate into the tube using 200-ul wide bore tips (FX-255-WB-R, Corning Axygen).
3   Rotate plates 180 degrees and repeat collection.
4   Visually verify that spheres are collected and manually collect left-over spheres.

#### Cisplatin barcoding

Monoisotopic cisplatin was used both as an orthogonal readout for distance to border (Durand, 1982) (Part physiology, Pt194) and to extend the 120-well barcoding to 240 wells by using Pt194 and Pt198 (part overexpression).

1   Wash pooled spheres with 4 ml PBS, centrifuge 1 min at 100 × *g* after each wash, and remove supernatant.

2    Remove supernatant and add 1 µM monoisotopic cisplatin in 1 ml PBS.

3    Incubate for 40-min shaking at 200 RPM.

4    Wash twice with CSM.

### Embedding

#### Preparation gelatine

1    Let 12% gelatine (Dr Oetker) swell in 0.1 M phosphate buffer (PB, pH 7.4) for 10 min.

2    Stir at 60°C for 4–6 h to dissolve.

3    Cool to 40°C

4    Add 2 µl/ml 10% sodium azide (Merck).

5    Keep at 37°C until use.

#### Preparation embedding mold

1    Get a clean glass rod with a flat bottom ca 3mm diameter and a conical top as an inverse mold (e.g., manufactured by a glassblower; see Fig EV1 for design).

2    Prepare 6% agarose (Invitrogen) in ddH20 by heating it in a microwave, keep at 80°C until use.

3    Pour hot agarose in 2-ml Eppendorf tube.

4    Insert inverted mold and put on ice for 10 min until agarose solidifies. Be careful to position the inverse mold exactly vertical, such that the flat bottom is horizontal.

5    Carefully remove inverse mold and wash cavity with PBS.

6    Prewarm to 37°C.

#### Sphere embedding

1    Incubate spheres for 5 min at 37°C.

2    Remove supernatant and add 4 ml warm gelatine and keep at 37°C for at least 10 min until spheres are sunk to the bottom.

3    Remove PBS from pre-warmed agarose mold and replace by warm gelatine.

4    Carefully transfer spheres using a 200-µl pipette with a wide bore (e.g., cut pipette tip). If not all spheres can be transferred at once, spin down agarose mold in pre-heated centrifuge (200 g, 37°C), remove supernatant gelatine, and transfer remaining spheres.

5    Use a pre-warmed 20-µl pipette and repeated spinning to carefully adjust position spheres, such that there is an even layer at the bottom of the cylindrical mold.

6    Let solidify the positioned spheres in the gelatine by incubating the agarose mold at 4°C overnight.

7    Carefully break the agarose mold to retrieve the gelatine plug. Hint 1: This is technically difficult. Train this step multiple times using an empty gelatine plug. In case the plug breaks apart at this step, it may be possible to re-melt the gelatine at 37°C and repeat the embedding. Hint 2: Instead of an agarose mold, a 4-ml sample tube with close-to-flat bottom or a flat-bottom tube could also be used for embedding.

8    Cryo-protect the gelatin plug by incubation for 1 h in 15% sucrose (Sigma) in ddH20 and then for 4h in 30% sucrose in ddH20 with 0.004% trypan blue (Sigma).

9    For cryo-embedding, prepare a cylindrical mold out of aluminum foil and fill it with OCT compound (Sakura).

10   Rinse plug with OCT compound and gently position it upright in the OCT mold, such that the sphere filled tip of the plug points upward. Hint: The superfluous gelatine from the plug can be trimmed.

11   Freeze in 40°C 2-methylbutane (Sigma).

12   Store frozen plug at −80°C.

### Cryo-sectioning

1    Mount the frozen plug on a cryo-microtome.

2    Cut slices (thickness: 6 µm, object temperature −17°C, knife temperature −15°C) and immediately melt them onto room temperature microscopy slides (Superfrost Plus, Thermo Scientific).

3    Dry the slides overnight at room temperature.

4    Image the sections using a bright-field microscope/slide scanner.

5    Store the slides at −80°C until usage.

### Antibody staining

1    Select sections with minimal tearing covering the whole volume of the plug.

2    Transfer sections from −80°C into TBS (50 mM Trizma base (Sigma), 50 mM NaCl (Sigma), pH 7.6).

3    Wash 3 times for 10 min with TBS.

4    Mark individual sections with a hydrophobic pen (Dako Pen, Agilent).

5    Block with 3% BSA in TBS-T (TBS + 0.1% Tween).

6    Prepare an antibody master mix in TBS and a final concentration of 1% BSA, 0.1% Tween, with antibody concentrations according to the panel.

7    A spillover slide was created for the whole panel by spotting ~0.3 µl antibody in 0.5 µl 0.4% trypan blue on an agarose-coated slide (Chevrier et al, 2018).

8    Remove the blocking buffer and add 12 µl antibody mix to each section.

9    Incubate overnight at 4°C in an hybridization chamber.

10   Wash the slides 3x in TBS for 10 min.

11   Add 20 µl of 1 µM Iridium Intercalator (Fluidigm) for 10 min.

12   Wash with TBS.

13   Add 1 µM Hoechst 33342 (Invitrogen) for 6 min.

14   Wash slides 3x with TBS for 10 min.

15   Dip slides in double-distilled water and blow dry immediately with compressed air.

16   Dry slides overnight in dark.

### Slide imaging

1    Image the dried slides using Axioscan Slide Scanner Z1 (Zeiss) using the DAPI (Hoechst) and the GFP channel, where appropriate.

2    Image slides using a Hyperion Imaging Mass Cytometer (Fluidigm) at nominal resolution of 1 $\mu m^2$ and an ablation frequency of 400 Hz.

### Cell line physiology experiment

Cells were seeded into the spheroid microplates at concentrations of 1×, 0.5×, and 0.25×, where the 1× concentrations were 3,200 cells per well for T-REx-293 cells, 6,400 cells per well for DLD-1 cells, 2,000 cells per well for T-47D cells, and 2,000 cells per well for HT-29 cells. Cells were grown in five replicates, and each plate was barcoded using a 60-well barcoding scheme. In plate p173, spheres of column 2, 3, 6, 7, 10, and 11 were incubated with 10 µM Telox 2 in 0.1% DMSO, other rows with 0.1% DMSO (control) for 4 h prior to fixation, and cells were fixed and barcoded after 72 h. The other plate, p176, was fixed and barcoded after 96 h. Monoisotopic cisplatin (194Pt, 1 µM) was added after pooling the spheroids. For

the 72-h time point, data were acquired on 18 cryo-sections, and for the 96-h time point, data were acquired on 16 cryo-sections.

### Chimeric overexpression experiments
#### Constructs
We used a library generated from the entry clones of a previously published cancer signaling constructs library (Martz *et al*, 2014). We added constructs encoding biologically relevant ligands and receptors from the human ORFeome V8.1 library (Dharmacon) via NEXUS Personalized Health Technologies at ETH Zurich (Yang *et al*, 2011). Destination vectors, including pDEST pcDNA5 FRT TO-eGFP, and pDEST 3' Triple Flag pcDNA5 FRT TO, were kindly provided by Anne-Claude Gingras (Lunenfeld-Tanenbaum Research Institute, Toronto, Canada (Couzens *et al*, 2013)). Tagged expression vectors were generated via Gateway Cloning (Invitrogen). End read Sanger sequencing was used to confirm the clone identity before transfection. Constructs were arranged on a master plate in a randomized fashion with control wells evenly distributed over the plate.

#### Experiment
T-REX 293 cells were seeded at a density of 20,000 cells per well in 100 µl medium into two 96-well flat-bottom cell culture plates (p155 and p156), using the normal medium prepared with tetracycline-free FBS (S182T-500, Biowest). After 24-h incubation, transfection was done using the jetPRIME transfection system (Polypus) according to the manufacturer's instructions: For each construct, a master-transfection mix of 22.5 µl jetPRIME buffer, 0.5 µl jetPRIME reagent, and 2.5 µl of 0.1 µg/µl DNA was prepared. An aliquot of 10 µl of this master mix was added dropwise to each well. After 5 h, the cell culture medium was changed using the Biomek Robot under semi-sterile conditions.

After 24 h, the cells were washed with PBS and detached by the addition of 100 µl 10× TrypLE Select Enzyme (Gibco) per well. Cells were resuspended in 100 µl medium. From these plates, cells were distributed into the spheroid microplates: From plate p155, 4 µl cell suspension per well was added to each well of plates p161 and p163. From plate p156, 20 µl of mock-transfected cells from border wells were transferred to each well, before 2 µl of the suspension was added per well to plates p165 and p167 and 4 µl was added per well to plates p169 and p171.

After 48 h, the spheres were imaged with bright-field microscopy. Subsequently, 2 µl of 50 µg/ml tetracycline hydrochloride (Sigma) in PBS to a final concentration of 1 µg/ml was added to each well. After 24 h, spheres were fixed and barcoded. For barcoding, three pairs of plates were barcoded using the 120-well barcoding plate layout (Plate 1: p161, p165, p171, Plate 2: p163, p167, p169). Then, plates p165 and p171 and plates p161 and 163 were pooled, and cisplatin (194Pt) was added. Plates p167 and p169 were pooled, and cisplatin 198Pt was added. Finally, plates p165, p171, p167, and p169 were pooled into one spheroid plug with 240 wells. p161 and p163 were embedded as a spheroid plug of 120 wells. After sectioning, 20 slices of the 120-well plug and 48 slices of the 240-well plug were selected for staining.

### Analysis
The computational analysis was implemented as a Snakemake workflow (Köster & Rahmann, 2018) using singularity containers (Kurtzer *et al*, 2017).

#### Spheroid diameter determination
In order to robustly determine spheroid diameter, we used a pipeline based on supervised pixel classification by Ilastik (Berg *et al*, 2019); this process identified spheres despite intensity variations. We used CellProfiler for segmentation and quantification (McQuin *et al*, 2018).

As quality control, bright-field images of each well were manually screened for spheroids with growth defects, such as particle or fiber inclusions, blinded for the spheroid growth condition.

#### IMC image analysis
Image processing of IMC data was based on our "imctools" library to convert raw IMC data to tiff files (Zanotelli *et al*, 2020a), custom the CellProfiler plugins "ImcPluginsCP" (Zanotelli *et al*, 2020b), and roughly followed the concepts laid out in our "ImcSegmentationPipeline" (Zanotelli & Bodenmiller 2017).

#### Quantification
To robustly identify spheroids in IMC images, we used supervised pixel classification by Ilastik (Berg *et al*, 2019) to identify spheroid centers, borders, and background, and used CellProfiler (McQuin *et al*, 2018) to segment the resulting probability maps. To identify cells, we used a similar approach, using Ilastik to classify pixels into nuclear/cytoplasmic membrane and background and CellProfiler to identify cells based on the resulting probability maps. Within cell regions, we quantified marker levels, applied compensation (Chevrier *et al*, 2018), and calculated other spatial features. Neighbors were identified by expanding each cell object by 3 pixels and identifying touching cells. We built an analysis framework ("spherpro"), based on Python (Van Rossum & Drake, 2009), SQLite, and Anndata (Wolf *et al*, 2018) to handle and analyze the data.

#### Image alignment
Fluorescent SlideScan images and IMC acquisitions were aligned with a fully automated iterative alignment process using TrakEM2 (Cardona *et al*, 2012). For the SlideScan images, the DAPI channel was used, and for the IMC acquisitions, the iridium channel was used.

First, whole spheroid plug sections were coarsely aligned using a rigid transform estimated from the machine provided global coordinate system. Then, the sections were aligned using a rigid alignment estimated by TrakEM2. Finally, individual cropped spheroid section images of the two modalities were fine-aligned using rigid alignment by TrakEM2.

#### Quality control
On a cellular segmentation level, several quality control criteria were applied:
- Sphere membership: At least 95% of all pixels of a cell need to be within the sphere segmentation region.
- Sphere ambiguous cells: Cells closer than 20 pixels to any other sphere in the image were excluded as they might be ambiguous.
- Cell size: Cells smaller than 10 pixels were excluded.
- Border cells: Cells directly touching the outer spheroid segmentation border were excluded.
- Main sphere: Cells not belonging to the largest contiguous cell mass of still valid cells were excluded.

- Fold classifier: A pixel classifier was trained in Ilastik based on DAPI fluorescence to identify folded areas in the spheres.

On a sphere slice (image) level, we used the following criteria to exclude images with all their cells:

- Not small: sphere slices with less than 10 valid cells (based on cellular segmentation quality control) were excluded.
- Manual QC: Quality control image consisting of raw images, quantification, spheroid and cell segmentation of DNA (193Ir), Histone H3, and 194Pt, and 198Pt channels were visualized for each spheroid slice with an anonymized ID. This allowed the visual identification of image artifacts such as folds, bubbles, tissue tearing, and spheroid mis-segmentations. Images were analyzed blind to spheroid growth conditions.

Additionally, we used bright-field images to identify wells with particles, fibers, deformed spheres, and multiple spheres. Images were analyzed blind to spheroid growth conditions. Sphere slices from these wells were excluded.

The effects of the individual QC steps on the datasets are summarized in Dataset EV2.

### Data transformation

If not otherwise indicated, the spillover-compensated mean pixel intensity per cell area was used as a readout. The data were log10 (x + 0.1)-transformed and winsorized using the 0.1$^{\text{th}}$ percentile.

### Debarcoding

For debarcoding, cells within 30 pixels from the outer spheroid border were considered. Over all images belonging to a spheroid plug, each barcoding channel was binarized with the average barcode channel intensity. Then, for each spheroid, the number of valid barcodes was determined. Due to the robustness of the barcode schemes used, false positives were infrequent. Sphere slices were assigned to the most common valid barcode in the slice. As quality control, at least 10 cells were required to be assigned to this barcode, and the most common barcode was required to be at least twice as frequent and then the second highest barcode. Images without valid identification were excluded from further analysis.

### Distance-to-border correction

Distance to the border of the spheroid slices can overestimate distance to border in the spheroid, as they represent spherical segments at different heights of the sphere. As the real spheroid diameter can be estimated by bright-field imaging, and assuming that spheroids are indeed spherical, this was correcting by the formula:

$$r_{real} = R - \sqrt{R^2 - 2rx + x^2}$$

where $r_{\text{real}}$ is the real distance to the sphere border, $R$ is the radius of the sphere measured in the bright-field images, $r$ is the radius of the segment, and $x$ is the measured (noncorrected) distance to the border in the segment.

### UMAP and cluster analysis

Uniform Manifold Approximation and Projection (UMAP) (McInnes et al, 2018) and clustering via the Leiden algorithm (Traag et al, 2019) were performed via SCANPY (Wolf et al, 2018).

### Marker variability analysis

For the marker variability analysis, the level of each marker ($y_p$) was predicted by a linear model (Fig 3A):

$$y_p = \beta_{pi} + BS(x_{d2border}) + \left(\sum_{m \neq p} \beta_m^{nb} x_m^{nb}\right) + \left(\beta_p^{nb} x_p^{nb}\right) + \left(\sum_{m \neq p} \beta_m^{\text{int}} x_m^{\text{int}}\right) + \in_{pi}$$

where:

- $y_p$: the level of marker $p$ in a cell.
- $\beta_{pi}$: technical staining/batch effect for marker $p$ of the image $i$ that the cell is part of
- $BS(x_{d2border})$: a nonlinear function of distance to border ($x_{d2border}$: represented by a polynomial B-spline of degree 3 with 10 knots distributed located at the deciles (10 quantiles).
- $x_m^{nb}$: average levels of marker $m$ in direct neighboring cells.
- $x_p^{nb}$: average levels of predicted marker $p$ in direct neighboring cells.
- $x_m^{\text{int}}$: cell internal marker levels of marker $m$.

The models and submodels were fitted using the statsmodels library (Seabold & Perktold, 2010).

If not mentioned otherwise, the reported variability explained ($R2$) for each model was the adjusted $R2$ relative to the adjusted $R2_{\text{tech}}$ the $R2$ of a model only containing an image-specific intercept. This prevented variability in signal differences resulting from technical issues (e.g., due to staining or acquisition) from being attributed to biological variability. This corrected $R2$ is calculated according to:

$$R2 = 1 - \left(\frac{1 - R2_{uncorr}}{1 - R2_{tech}}\right).$$

For the correlation heat maps and distance-to-border plots (Figs 2C and EV4), an image-specific intercept was fit. This intercept was subtracted before calculating the Pearson correlation.

### Permutation analysis

We fit all possible sequences of adding the modules for global environment ($BS(x_{d2border})$), local environment ($x_m^{nb}$), autocorrelation ($x_p^{nb}$), and cell state ($x_m^{\text{int}}$) to the model, for each marker and condition, and recorded the additional marker variability explained at each step in each sequence. For each sequence, we calculated the variance of the additional variability explained by each added module. Given that the total variability explained is independent of the sequence, high variance suggests that marker variability is explained by a few modules and low variance suggests that marker variability is explained by multiple modules.

In a strictly hierarchical dependency structure, modules higher in the hierarchy should contain the variability explained by those lower in the hierarchy. Adding a more explanatory module before a less explanatory one will lead to the former fully capturing the variability of the latter, yielding high variance. Sequentially fitting modules in line with the hierarchy of explanatory power should maximize the contributions of each module, thus reducing the variance. The optimal order corresponds to the sequence with least variance.

### Chimeric overexpression analysis

We trained a pixel classifier based on the two IMC GFP antibodies to robustly detect overexpressing image regions in which a

construct was overexpressed. We calculated the average pixel-wise probability for overexpression and required this to be more than 0.01 (estimated false discovery rate: 0.003) for a cell to be classified as "overexpressing". Cells with an average pixel-wise probability higher than 0.01 but lower than 30% of the maximal value observed in neighboring cells were added to an "ambiguous" category. Other cells within 6 pixels of "overexpressing" cells were classified as "neighbor" cells. All other cells were classified as "bystanders".

This classification was not reliably possible for spheres transfected with a FLAG-only construct without GFP, due to FLAG antibody background staining. All cells in such spheres were classified as "bystander" cells.

We used a linear mixed-effects model to estimate marker levels independently of the effect of belonging to the overexpressing, neighboring, or bystander cell class of a specific construct. The following model was used to predict a given marker level $y_p$:

$$y_p = BS(x_{d2border}) + \beta_{(ctrl|oexp|nb)*construct} + \beta_{plateid} + (1|spheroid) + (1|siteid) + (1|imageid) + \in_p.$$

where:

- $BS(x_{d2border})$: a nonlinear function of distance to border ($x_{d2border}$: represented by a polynomial B-spline of degree 3 with 10 knots distributed located at the deciles (10 quantiles)
- $\beta_{(ctrl|oexp|nb)}$+$construct$: an intercept for combination of construct and overexpression class
- $\beta_{plateid}$: a fixed effect intercept for belonging to any of the 6 plates. This accounts for plate-wise effects.
- $(1|spheroid)$: a random effect acknowledging that cells from the same sphere are correlated
- $(1|siteid)$: a random effect acknowledging that sphere sections/images that were stained and acquired together are not independent, e.g., through staining effects
- $(1|imageid)$: a random effect acknowledging that cells from the same sphere slide/image are not independent.
- $\epsilon$: residual variation. This is assumed to be homoscedasticity.

## Data availability

The datasets and computer code produced in this study are available in the following databases:

- Raw imaging data: Zenodo Record 4055781 (https://zenodo.org/record/4055781)
- Code to reproduce the analysis from raw data: Zenodo Record 4071862 (https://zenodo.org/record/4071862) / GitHub (https://github.com/BodenmillerGroup/SpheroidPublication)

**Expanded View** for this article is available online.

## Acknowledgements

The authors acknowledge the assistance and support of the Center for Microscopy and Image Analysis, University of Zurich. We acknowledge in particular José María Mateos Melero and Claudia Meyer from the Institute of Anatomy, University of Zurich, for their expertise and equipment for histological embedding and cutting. We are also in debt to Min Lu and Kris C. Wood for sharing the entry clones for the cancer-related signaling constructs library. We are also grateful to all contributors to the open-source software packages that made this project feasible. Finally, we would like to thank all members of the Bodenmiller and Gerber Labs for their support and input. In particular, we would like to thank Artur Yakimovich and Vardan Andriasyan for their expertise with spheroid culture and imaging. BB's research was supported by a SNSF Assistant Professorship Grant, an NIH Grant (UC4 DK108132), and by the European Research Council (ERC) under the European Union's Seventh Framework Program (FP/2007-2013)/ERC Grant Agreement No. 336921.

## Author contributions

VZ and BB conceptualized this study. VZ and ML developed the Wet Lab and computational workflow with support from BB, XL, and FG. VZ analyzed the data and generated the figures with feedback from BB and NS. VZ, NS, and BB wrote the manuscript, and XL, FG, and ML provided feedback. BB acquired funding.

## Conflict of interest

The authors declare that they have no conflict of interest.

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
