## [Review Process File · Molecular Systems Biology]

A quantitative analysis of the interplay of environment, neighborhood and cell state in 3D spheroids

Vito Riccardo Tomaso Zanotelli, Matthias Leutenegger, Xiao-Kang Lun, Fanny Georgi, Natalie de Souza, and Bernd Bodenmiller

DOI: [10.15252/msb.20209798](https://doi.org/10.15252/msb.20209798)

Corresponding author(s): Bernd Bodenmiller (bernd.bodenmiller@uzh.ch)

Review Timeline:

Submission Date:	20th Jun 20
Editorial Decision:	4th Aug 20
Revision Received:	21st Oct 20
Editorial Decision:	27th Oct 20
Revision Received:	11th Nov 20
Accepted:	12th Nov 20

Editor: Maria Polychronidou

Transaction Report:

Thank you again for submitting your work to Molecular Systems Biology. We have now heard back from the three referees who agreed to evaluate your study. Overall, the reviewers are rather supportive. However, they raise a series of concerns, which we would ask you to address in a revision.

I think that the recommendations of the reviewers are rather clear and there is therefore no need to repeat the points listed below. Most issues raised are relatively minor and refer to the need to improve the data presentation and the clarity of the text. Reviewers #2 and #3 are somewhat concerned that the extent of biological insight provided by the study comes across as limited. To address this point we would ask you to make sure that the core biological conclusions/implications are clearly described in the text (even if that is mainly in the Discussion). Please let me know in case you would like to discuss in further detail any of the issues raised.

On a more editorial level, we would ask you to address the following issues.

Reviewer #1:

Here Bodenmiller and colleagues perform an analysis of cell state activity considering single cells, local neighbourhood, and global positioning in cancer spheroids. Through rigorous analysis they find that single cell state is largely predicted by external factors.

This fascinating study realizes experiments many researchers conceive of doing. This could have only be done using the sophisticated methodologies the authors have developed. The authors should be applauded for both generating this dataset, and performing very rigorous analysis of these data.

I have very little issue with this work. It could likely be published as is.

As a major criticism I find some of the sections (i.e. description of Figures 3 and Figure 4) somewhat difficult to follow - and I would consider myself an expert in the field. Thus a general reader might find them challenging to get through. It might be worth working on making these sections more clear because I think the findings here are quite important, and will be interesting to a wide variety of scientists.

There are two aspects to this. 1) The statistical analysis that is being done is difficult to understand. Perhaps the addition of specific/clear example scenarios and outcomes would help. 2) The result (and implications) of the analysis are also not always clear. Here, the addition of some concluding statements at the end of each paragraph, which make the statistical finding made by the relevant analysis understandable to a 'lay' cell biologist, would be very helpful.

As a very minor criticism the authors seem intent on having this work presented largely as a resource. This is fine. But there probably is some biology in here that might be worth discussing.

Reviewer #2:

Summary

Zanotelli et al describe an impressive high-throughput and high-dimensional IMC method to study mono-culture spheroids. They apply this technology to x4 mono-culture cell line spheroids to observe a signaling hierarchy comprising global environmental, local neighbourhood, and cell state features. The authors then overexpress various signaling cues to explore their effect on mono-culture cell-cell communication. The work is generally of high quality - although sometimes the clarity of the biological conclusions are lost amongst the data.

General Remarks

The technical quality of the science throughout the manuscript is high. The barcoding > pooled plug

> stain > IMC approach is particularly novel and will be of high interest to many labs working with 3D cultures. If anything, I wish the manuscript included more detail on the methodological advances presented in this paper!

The 3D in vitro culture systems are simple by modern standards and unlikely to mimic many in vivo biological processes. However, rather than over-extrapolate from these limited models, the authors

leverage the uniformity of spheroids to derive gradient-based biological mechanisms. The authors also use a clever overexpression system to test local non-cell autonomous signaling. This cellular system is potentially more representative of a heterogeneous tumor microenvironment.

This reviewer is not qualified to critique the image analysis components of this paper and I leave that to other reviewers.

The manuscript could be greatly improved with some adjustments to figure legibility. It is often difficult to understand what is being shown in each figure. Some suggestions can be found in the 'Minor Points' section below. The paper also sometimes reads as a list of percentages going up and down. Focus should really be given to the biological conclusions being drawn - not just the numbers. After reading the paper, I found it hard to summarise the core biological findings despite the high-quality of the underlying science.

Overall the paper is solid and if the authors can improve the readability and address some additional points below, the paper is worthy of publication in MSB. I congratulate the authors on a very interesting piece of work.

Major Points

1) It was not clear to me if all spheroid cross-sections are taken across the centre of each spheroid. This could have major implications on the biological processes captured within each spheroid section and therefore the underlying data analysis. For example, for two spheroids of identical geometry, a section through the middle of the spheroid would contain more hypoxic cells than a section that only cut through the 'top' 5% (i.e. containing cells largely on the edge of the spheroid). In both scenarios, a circular section would be formed in 2D. How can the authors guarantee that all spheroids are sectioned in a comparable way? If not, how could this be improved in future versions of the technology?

Minor Points

1) Fig.1. The barcoding and sample-processing methods described in this paper will be of major interest to readers. However, Fig. 1 which describes this workflow, is sparsely labeled and difficult to follow without constant referral to the text. Some suggestions to improve this for readers:

- Fig. 1a: the spheroid diagrams should be labeled and aligned with the model to more clearly illustrate the biological problem being addressed.
- Fig. 1b: parameters could be coloured by marker class? Cell signaling, cell cycle / cell state, varia?
- Fig. 1c: the spheroid formation cartoon is confusing, not related to the downstream analysis, and should probably be removed. Instead it would be useful to label the brightfield matrix with some hypothetical variables (e.g. cell-types, time points, overexpressions etc) so readers can appreciate the scale of the experiments. The switch from coloured barcoding plate to pooled plug also needs to be properly explained. This is a major innovation from the paper and just breezed over. It's really clever - show it off! What section of the plug is cut? The side-view makes it look like the spheroids do not occupy a uniform layer (see Major Point above). 'Image Quantification' and 'Data Analysis' also sound very similar even though they're very different processes in the workflow. More detail would be useful here. What type of quantification and analysis are being done? When is computation performed on the plug and when on the local spheroid? A general lack of detail throughout Fig. 1 undersells the quality of this work.

- 2) The antibody panel (Table S1) should include the metal conjugate used for each antibody.
- 3) 't-Protein' is not a typical annotation for total protein levels and confused me on first reading. I kept thinking it referred to a threonine modification or some unusual PTM. I would advise using the following syntax: 'p-Protein [site]' for all phospho modifications, 'c-Protein [site]' for protease cleavage, and simply leaving antibodies against unmodified epitopes as 'Protein'.
- 4) It's not immediately obvious if the reader is looking at IMC images of a plug or a single spheroid. Both are circles with smaller circles inside! To aid the reader I suggest labeling the top row of Fig. 2a with 'Plug (# No. Spheroids)' and the bottom row with 'Spheroid' in the figure directly, not just the legend. Also, which spheroid is being shown in the bottom row? Can you draw an arrow to show the reader? This would help solidify the relationship between the plug and the single spheroid.
- 5) From what I can tell the correlations in Fig. 2c (cell-state) are symmetrical. If so, it would be clearer for readers to only show one half of the heatmap matrix.
- 6) Can the clusters in Fig. 2c be given a broad biological name? This would help readers better understand the figure.
- 7) The different components of the model in Fig. 3a, c, d, e should be labeled. Explanations in the figure legend alone are not enough to understand this figure quickly.
- 8) Cell-state is a subjective term and used differently by different labs. For some it means differentiation state, others it's cell-cycle / apoptosis. The authors should define what they mean by cell-state and how it relates to the cell-cycle class also studied.
- 9) c-Caspase 3 does not correlate with c-PARP. Why?
- 10) The overexpression system in Fig. 5 is very clever and plays to the spatial power of IMC vs traditional CyTOF. The authors show that overexpression of cues regulates cell-autonomous signaling, and in some cases, also cell-non-autonomous signaling. Are there any examples of reciprocal signaling? For example, when a GFP+ overexpressing cell has a different signaling profile only when overexpressing cell neighbours are also altered? Such events are probably quite rare in mono-cultures (where neighbour cells are quite uniform), but would potentially quite common in co-cultures. If reciprocal signaling is not present in this dataset, it should be commented in the discussion as a feature that should be found in co-cultures.

Reviewer #3:

Dear Editor, dear Authors,

the manuscript MSB-20-9798 entitled 'A quantitative analysis of the interplay of environment, neighborhood and cell state in 3D spheroids' by Zanutelli and colleagues used their antibody-based multiplexed imaging mass cytometry (IMC) pipeline (ref. to 17-19) with a panel of 34 antibody targets to study epithelia cell line multicellular spheroids (including four cell lines at different time points and growth conditions). The authors try to dissect or model global, neighborhood and cell states as different independent interacting modules. In principle, the goal is to access and explain spheroid heterogeneity or phenotypes through marker variability analysis. To further validate an expected strong cell-cell interaction component (see summary 3.2.1), a gene overexpression assay

was performed for biological conformation. In summary, again as expected (summary 3.2), the authors conclude from these results that intracellular expressions 'are closely related to the cellular states of neighbors and the spatial location of cells in the global environment'.

The selection of markers for this experiment clearly biases the modeling outcome. More than half of the proteins are mitotic or cell-cycle related, therefore a linear model actually would explain 'on average half of the variability' (see abstract), when cells enter specific cell cycle states, i.e. these observed pathways are coordinated.

How useful or artificial are these 'modules' of intracellular, cellular, local, global in spheroids at all, if they are closely related? And what do we learn from these comparisons to explain e.g. cancer (see motivation in introduction)? Especially, knowing that cancer does not evolve monoclonal, nor cell lines are monogenic (<https://dmm.biologists.org/content/11/11/dmm037366>)? Cell lines are polyclonal and are composed of different cell states and types, which means that derived multicellular tumorspheroids are by definition heterogenous. The current manuscript does not introduce, model nor discuss this aspect.

Therefore, above 'findings' might be not valid anymore, if tumor cell type 'mixtures' would be used with strong inter-spheroid heterogeneity (also in time). In the context of known and shown inter-spheroid heterogeneity (referring to Fig. 2a, example IMC images of a spheroid plug - HT29 only -, top row), the manuscript is not informative or 'tangible'. The assumption of basic understanding disease phenotypic patterns within spheroids by established cell lines might be not given, because malignant or patient-derived material would display different scenarios using more appropriate marker panels (e.g. epigenetic/developmental).

Fig.2c should have a clear separation by headers. Not sure if Fig. 1a and Fig. 3a, c, d, e schemes are helpful for understanding of the points/models made/computed.

Despite the excellent technical and analytical methods presented in the manuscript, the drawn interpretations are not very biological insightful or novel, while the fundamental question of biological heterogeneity and wired interplay cannot be addressed due to the experimental design. In conclusion, the technical quality would justify publication of the manuscript, while biological application/findings and relevance might not.

Editor:

Thank you again for submitting your work to Molecular Systems Biology. We have now heard back from the three referees who agreed to evaluate your study. Overall, the reviewers are rather supportive. However, they raise a series of concerns, which we would ask you to address in a revision.

I think that the recommendations of the reviewers are rather clear and there is therefore no need to repeat the points listed below. Most issues raised are relatively minor and refer to the need to improve the data presentation and the clarity of the text. Reviewers #2 and #3 are somewhat concerned that the extent of biological insight provided by the study comes across as limited. To address this point we would ask you to make sure that the core biological conclusions/implications are clearly described in the text (even if that is mainly in the Discussion). Please let me know in case you would like to discuss in further detail any of the issues raised.

Reply to the editor:

We first would like to thank the editor for organizing the review process, and the three reviewers for their encouraging feedback and constructive comments. Guided by the reviewers' recommendations, we have made changes to text and figures to highlight biological examples and to improve the clarity of our message. We hope that all comments and concerns satisfactorily.

We would like to point out one minor change to the data. As part of the revision we tried to better document our data processing pipeline. During this process we noticed that we had quantified the distance-to-border of cells in a slightly sub-optimal way: Instead of using the average distance of all pixels of a cell to the border, we took the maximal distance of any pixel. Since this is likely to be biologically less meaningful, we repeated our analyses using the average distance. With this new analysis, the global environment explained slightly more of the marker variation (median variation explained before: 7.6%, now 8.0%). This subtly affected graphs and exact numbers and did not change our main messages. Only two minor conclusions were affected:

The first exception was the comparison shown in the original Figure 5C. Here we had stated the following in the initial version of the manuscript:

"We found that the set of markers (57% of all markers) that were significantly perturbed in either neighbors or bystanders in any of the overexpression experiments indeed had a slightly higher dependence on neighboring cell marker levels in our analysis of unperturbed T-REx-293 spheres grown for 96 hours ($R3_{\text{global+local}} - R3_{\text{global}}=4\%$ vs. 5%, t-test $p = 0.03$, Kruskal–Wallis $p=0.05$, $n=34$, Fig. 5e)"

After re-analysis with the better distance-to-border estimate, we found that although the data only slightly changed (Fig R1), this relationship became non-significant (t-test $p = 0.11$, Kruskal–Wallis $p=0.22$). This was a minor aspect of the manuscript, and a weak effect to start with, so we have removed this observation from the revised manuscript.

Figure R1: Each dot represents the average dependence of markers on the local neighborhood in unperturbed T-REx-293 spheres grown for 96 hours after subtracting the dependence on the global environment. Markers significantly affected by any overexpression in neighboring cells (True) show on average a higher dependence than markers not affected by any overexpression. A) Analysis reported in first submission B) Re-analysis with improved distance-to-border estimate.

Second, we note that due to minor numerical differences in the complete re-analysis of all our data, the order of clusters - but not the clusters themselves - changed in the clustered heatmaps (Fig. 2C).

Reviewer #1:

Here Bodenmiller and colleagues perform an analysis of cell state activity considering single cells, local neighbourhood, and global positioning in cancer spheroids. Through rigorous analysis they find that single cell state is largely predicted by external factors.

This fascinating study realizes experiments many researchers conceive of doing. This could have only be done using the sophisticated methodologies the authors have developed. The authors should be applauded for both generating this dataset, and performing very rigorous analysis of these data.

I have very little issue with this work. It could likely be published as is.

As a major criticism I find some of the sections (i.e. description of Figures 3 and Figure 4) somewhat difficult to follow - and I would consider myself an expert in the field. Thus a general reader might find them challenging to get through. It might be worth working on making these sections more clear because I think the findings here are quite important, and will be interesting to a wide variety of scientists.

There are two aspects to this

1) The statistical analysis that is being done is difficult to understand. Perhaps the addition of specific/clear example scenarios and outcomes would help.

We would like to thank the reviewer for these good suggestions. We now illustrate patterns of explanatory power for variation of three specific markers (carbonic anhydrase, p-S6, p-Rb) as the different modules (i.e., internal cell state, autocorrelation, local neighbourhood and global environment) are added to the model. This is shown in Figure EV5 and we have added a discussion of the consequences of assuming independence of the different modules to the main text. We hope that clarifies our point that not accounting for the interdependency of factors that influence cellular phenotype can result in important effects being missed or misinterpreted.

2) The result (and implications) of the analysis are also not always clear. Here, the addition of some concluding statements at the end of each paragraph, which make the statistical finding made by the relevant analysis understandable to a 'lay' cell biologist, would be very helpful.

We hope the examples and discussion we have added in response to point 1 will make the analysis more understandable. Further we also added additional concluding and summary statements to make our main points more accessible.

As a very minor criticism the authors seem intent on having this work presented largely as a resource. This is fine. But there probably is some biology in here that might be worth discussing.

We agree that there are likely to be specific biological points that could be of interest to other researchers and we hope that the dataset does indeed prove a rich resource. The main goal of the described study was to use the homogenous spheroid model system to elucidate systems-level principles about the factors that influence single-cell phenotypic variability in 3D tissues. The key conceptual point that emerged from our study is that cell-intrinsic and environmental factors do not affect marker variability in an independent fashion. This interdependency must be accounted for when drawing conclusions about factors that influence cell phenotype. In our revised manuscript, we have tried to bring across this point more clearly, using specific examples as suggested by the referee. We decided not to add much discussion of isolated biological conclusions to the manuscript since toggling between such results and the systems-level concepts we want to convey would likely confuse readers.

Reviewer #2:

Summary

Zanotelli et al describe an impressive high-throughput and high-dimensional IMC method to study mono-culture spheroids. They apply this technology to x4 mono-culture cell line spheroids to observe a signaling hierarchy comprising global environmental, local neighbourhood, and cell state features. The authors then overexpress various signaling cues to explore their effect on mono-culture cell-cell communication. The work is generally of high quality - although sometimes the clarity of the biological conclusions are lost amongst the data.

General Remarks

The technical quality of the science throughout the manuscript is high. The barcoding > pooled plug > stain > IMC approach is particularly novel and will be of high interest to many labs working with 3D cultures. If anything, I wish the manuscript included more detail on the methodological advances presented in this paper!

We have now added much more methodological detail in the form of the structured methods. We also added Figure EV1, which illustrates the full experimental and analytical workflow. We have also tried to highlight the specific methodological advances better within the main text.

The 3D in vitro culture systems are simple by modern standards and unlikely to mimic many in vivo biological processes. However, rather than over-extrapolate from these limited models, the authors leverage the uniformity of spheroids to derive gradient-based biological mechanisms. The authors also use a clever overexpression system to test local non-cell autonomous signaling. This cellular system is potentially more representative of a heterogenous tumor microenvironment.

This reviewer is not qualified to critique the image analysis components of this paper and I leave that to other reviewers.

The manuscript could be greatly improved with some adjustments to figure legibility. It is often difficult to understand what is being shown in each figure. Some suggestions can be found in the 'Minor Points' section below. The paper also sometimes reads as a list of percentages going up and down. Focus should really be given to the biological conclusions being drawn - not just the numbers. After reading the paper, I found it hard to summarise the core biological findings despite the high-quality of the underlying science.

In response to these comments, we have revised sections of the manuscript to better emphasize the main biological message we want to convey, rather than focusing too much on the quantitative aspects. We note that, in some cases, the quantification is key, and we have left the numbers in the main text. We hope that the reviewers find that we have struck a better balance in this revised version of our manuscript.

Our main biological messages are largely conceptual that cell-intrinsic and environmental factors do not affect marker variability in an independent fashion and that this interdependency should be accounted for when drawing conclusions about factors that influence cell phenotype (see also our answer to reviewer #1). In our revised manuscript, we now use specific biological examples to better bring across this point and to make it more intuitive for biologists. Further, given the homogeneity of the model system, specific biological findings may not be translatable.

Overall the paper is solid and if the authors can improve the readability and address some additional points below, the paper is worthy of publication in MSB. I congratulate the authors on a very interesting piece of work.

Major Points

1) It was not clear to me if all spheroid cross-sections are taken across the centre of each spheroid. This could have major implications on the biological processes captured within each spheroid section and therefore the underlying data analysis. For example, for two spheroids of identical geometry, a section through the middle of the spheroid would contain more hypoxic cells than a section that only cut through the 'top' 5% (i.e. containing cells largely on the edge of the spheroid). In both scenarios, a circular section would be formed in 2D. How can the authors guarantee that all spheroids are sectioned in a comparable way?

We measured random sections through the spheres (average 5-7 sections/sphere). Thus our analyses never relied on single but multiple sections, which should better represent the overall sphere than individual sections. Further, our analysis of marker variability was usually done on the level of sphere growth condition, thus on relying on several sections from several spheres. Notably, even when performed on a per-sphere basis, we find that our marker variability analysis was reproducible (see for example Fig EV5B, overall SD < 0.04). This suggests that we sampled a sufficient number of sections to representatively capture overall sphere variability from individual sphere sections. We have revised the workflow illustrated in Figure 1 as well as the text to make it clearer that we indeed had multiple, random sections per sphere.

In the case of the overexpression analysis, we use a linear mixed effects model to account for the fact that slices are random samples of spheres and that cells in slices are not independent observations.

We further note that we corrected our distance-to-border estimate for the fact that we take random - and thus sometimes tangential - sections, which leads to a systematic overestimation of distance-to-border when naively taking the distance-to-mask border as an estimate. We measured the diameter of the intact sphere with brightfield imaging, measured the diameter of the section in the mass cytometry image, and - assuming sphericity - used simple geometry to estimate the 'corrected' distance-to-border estimate. This is described in detail in the Methods. We also verified that such a distance estimate correlates well with a nonspecifically binding reagent that we let diffuse into the spheres (CisPt) - that provided an orthogonal readout of distance-to-border (Durand, 1982). In our 4 cell line dataset, this correction significantly improved the Spearman correlation (ρ) of CisPt levels and distance-to-border in 67% of the spheroids by an average of $\mu(\Delta\rho)=-0.010$ from $\mu(\rho_{\text{uncorr}})=-0.905$ to $\mu(\rho_{\text{corr}})=-0.915$ ($n=100$ spheres, Significance: t-test for $\mu(\Delta\rho)=0$: $p=0.004$, bootstrapped p-value for $\mu(\Delta\rho)\Rightarrow 0$: $p=0.002$ (100,000 samples)). The Spearman correlation of distance to border with CisPt over all cells from all conditions and cell lines from this dataset improved from -0.8 to -0.82 (Figure R2B,C). We now mention in the methods that CisPt can be used as a diffusion stain according to (Durand, 1982) and have added a plot showing this correlation here (Figure R2).

Figure R2:

A) Left: Spearman correlation of average cellular ^{194}Pt mean pixel intensity and uncorrected distance-to-border estimate per sphere.
 Middle: Spearman correlation of ^{194}Pt and corrected distance-to-border estimate.
 Right: histogram of the delta of the two estimates. B-C) Heatplot of raw (B) and corrected (C) distance-to-border with average CisPt intensity per cell. Color indicates point density. All cells from the four cell line dataset were used.

If not, how could this be improved in future versions of the technology?

Based on our proof of concept data it may be feasible to coarsely align in 3D the slidescans retrieved after cutting the block (Figure R3). Combining such a coarse registration with spheroid level segmentation would allow selection of sections that contain particular cuts of interest, e.g. center cuts, before the expensive staining and IMC ablation step. We also envision that future iterations of the IMC technology will enable complete 3D analysis of all studied spheres (Catena *et al*, 2020).

Figure R3: A) Rendering of a proof-of-concept 3D alignment of 93 cuts containing 120 spheres (spheroid plug p165) based on DAPI fluorescent slidescan images aligned using TrakEM2. B) Section along X-Z axis. C) Section along Y-Z axis.

Minor Points

1) Fig.1. The barcoding and sample-processing methods described in this paper will be of major interest to readers. However, Fig. 1 which describes this workflow, is sparsely labeled and difficult to follow without constant referral to the text. Some suggestions to improve this for readers:

- Fig. 1a: the spheroid diagrams should be labeled and aligned with the model to more clearly illustrate the biological problem being addressed.

We have better aligned the diagrams and the graphical model, and have added labels and legends to better explain individual elements.

- Fig. 1b: parameters could be coloured by marker class? Cell signaling, cell cycle / cell state, varia?

We added a colored background to indicate the marker classes.

- Fig. 1c: the spheroid formation cartoon is confusing, not related to the downstream analysis, and should probably be removed. Instead it would be useful to label the brightfield matrix with some hypothetical variables (e.g. cell-types, time points, overexpressions etc) so readers can appreciate the scale of the experiments.

We have removed the spheroid formation cartoon. We chose not to add hypothetical variable labels to the brightfield matrix as we are concerned this would have made the image too busy, but we can do so if it's still thought to be necessary.

The switch from coloured barcoding plate to pooled plug also needs to be properly explained. This is a major innovation from the paper and just breezed over. It's really clever - show it off! What section of the plug is cut? The side-view makes it look like the spheroids do not occupy a uniform layer (see Major Point above). 'Image Quantification' and 'Data Analysis' also sound very similar even though they're very different processes in the workflow. More detail would be useful here. What type of quantification and analysis are being done? When is computation performed on the plug and when on the local spheroid? A general lack of detail throughout Fig. 1 undersells the quality of this work.

We have added a graphic indicating the pooling step and now illustrate more clearly that many sections are cut through a plug. We indicate the number of sections (16), spheroids (283) and individual single cells (141'446) we analysed. We also added Figure EV1 which illustrates the workflow in detail.

'Image quantification' is the extraction of information in the form of tabular measurements from images. With 'Data analysis' we mean the project specific, statistical analysis of these extracted measurements and their relationship to the different perturbations used. This clarification has been added to the legend of Figure 1C.

2) The antibody panel (Table S1) should include the metal conjugate used for each antibody.

Thanks for pointing this out. The metal conjugation information indeed got lost during the preparation of the table. The revised table includes this information.

3) 't-Protein' is not a typical annotation for total protein levels and confused me on first reading. I kept thinking it referred to a threonine modification or some unusual PTM. I would advise using the following syntax: 'p-Protein [site]' for all phospho modifications, 'c-Protein [site]' for protease cleavage, and simply leaving antibodies against unmodified epitopes as 'Protein'.

We have made the suggested change.

4) It's not immediately obvious if the reader is looking at IMC images of a plug or a single spheroid. Both are circles with smaller circles inside! To aid the reader I suggest labeling the top row of Fig. 2a with 'Plug (# No. Spheroids)' and the bottom row with 'Spheroid' in the figure directly, not just the legend. Also, which spheroid is being shown in the bottom row? Can you draw an arrow to show the reader? This would help solidify the relationship between the plug and the single spheroid.

We added the suggested annotations.

5) From what I can tell the correlations in Fig. 2c (cell-state) are symmetrical. If so, it would be clearer for readers to only show one half of the heatmap matrix.

Figure 2C is indeed symmetrical. We prefer to show the full heatmap, as this facilitates labeling and annotations. We have added 'symmetrical' to the text to make it clear that the matrix is symmetrical.

6) Can the clusters in Fig. 2c be given a broad biological name? This would help readers better understand the figure.

Since the clusters do not represent 'pure' biological processes, we refrained from naming them. However, in the revised version we now indicated which markers characterise which clusters, such that a reader can more easily draw their own conclusions.

7) The different components of the model in Fig. 3a, c, d, e should be labeled. Explanations in the figure legend alone are not enough to understand this figure quickly.

We have added more annotations to these figures.

8) Cell-state is a subjective term and used differently by different labs. For some it means differentiation state, others it's cell-cycle / apoptosis. The authors should define what they mean by cell-state and how it relates to the cell-cycle class also studied.

We clarified that in this study we use 'cell state' to mean 'levels of intracellular markers' and made sure that it also consistently used this way.

9) c-Caspase 3 does not correlate with c-PARP. Why?

There are two reasons for the lack of correlation: First, apoptotic cells are very rare and are even entirely absent in some conditions (Figure R4). In spheroids formed from HT29 cells, we detected no apoptotic cells under the conditions used. Consequently both cleaved PARP and cleaved Caspase reflect background staining for this cell line and thus the lack of correlation is not surprising. For DLD1 spheroids, where apoptotic cells are more abundant, we did observe a good correlation (see Figure EV5b).

Figure R4. The fraction of apoptotic cells in spheroids of all four cell lines under the indicated seeding and growth conditions. Apoptotic cells were identified based on identification of apoptotic clusters, which we did by clustering (cell cycle markers + apoptotic markers).

Second, cleaved Caspase has higher and different background staining than cleaved PARP. In the scatterplot of cleaved PARP vs. cleaved Caspase, the population negative for cleaved PARP varies between a mean intensity of 0 and approximately 0.3 counts per pixel per cell ($\log_{10}(x+0.1) = (-1, -0.5)$), whereas cleaved Caspase background shows higher variation with a mean intensity ranging from 0 to 1 counts in the negative population. In this range the markers do not correlate, which was expected as there is no biological reason why the

background staining of these two markers should be correlated. In cells that are positive for these markers (mean intensity > 1 , $\log_{10}(x+0.1)=0$), both markers are strongly correlated (Figure R5).

Figure R5: Scatter plots of cleaved Caspase (x-axis) vs. cleaved PARP (y-axis) mean metal counts per cell colored by the point density. Panels show data from each of the four cell lines for all cells from spheres in the growth conditions with the largest sphere sizes.

10) The overexpression system in Fig. 5 is very clever and plays to the spatial power of IMC vs traditional CyTOF. The authors show that overexpression of cues regulates cell-autonomous signaling, and in some cases, also cell-non-autonomous signaling. Are there any examples of reciprocal signaling? For example, when a GFP+ overexpressing cell has a different signaling profile only when overexpressing cell neighbours are also altered? Such events are probably quite rare in mono-cultures (where neighbour cells are quite uniform), but would potentially quite common in co-cultures. If reciprocal signaling is not present in this dataset, it should be commented in the discussion as a feature that should be found in co-cultures.

This is an interesting question, and we looked for effects like this. For example, we asked if the average marker level varies as a function of both overexpression level in a cell and the maximal overexpression level in any neighbouring cell. We found that in some cases, we indeed observed that marker expression in an overexpressing cell depended on overexpression in neighbouring cells, which could be seen as reciprocal signaling.

We illustrate here the clearest example of such an effect (Figure R6): We evaluated whether the effect of overexpression of MEK1-DD on markers is affected by the overexpression levels of neighbours. We found that for some markers, for example pMEK1, the level increases strictly as a function of overexpression within a cell but is not dependent on the

overexpression level in the neighbouring cell. This results in a gradient and contour lines that are parallel to the y-axis. In contrast, for markers like pERK and pS6, the increase is greater in cells whose neighbours have lower levels of the overexpression construct (right slanted gradient/contour lines), indicating that there may be some reciprocal signaling where strong activation of MEK1-DD in neighbouring cells dampens the cell intrinsic effect of overexpression in a given cell. pFAK is representative of a marker that does not respond to this overexpression.

Fig R6: Results of overexpression of MEK1-DD (MEK-1 (S218D/S222D)-GFP). Each panel shows the average level of the indicated marker (color) after binning cells according to GFP levels of a cell (x axis) and the maximal observed GFP levels of any cell in the immediate neighbourhood (y axis, 15 bins each). Black lines indicate contour levels of binned intensity. Analysis averaged over all cells and spheres from the overexpression condition.

Although we definitely share your excitement for these types of analyses and agree that our overexpression data is very suitable, we think that a systematic assessment of such effects warrants a separate study. In particular, this type of analysis would require the development of new statistical tools to adequately capture such effects. We have added a comment about the potential to discover such phenomena to the Discussion.

Reviewer #3:

Dear Editor, dear Authors,

the manuscript MSB-20-9798 entitled 'A quantitative analysis of the interplay of environment, neighborhood and cell state in 3D spheroids' by Zanotelli and colleagues used their antibody-based multiplexed imaging mass cytometry (IMC) pipeline (ref. to 17-19) with a panel of 34 antibody targets to study epithelia cell line multicellular spheroids (including four cell lines at different time points and growth conditions). The authors try to dissect or model global, neighborhood and cell states as different independent interacting modules. In principle, the goal is to access and explain spheroid heterogeneity or phenotypes through marker variability analysis. To further validate an expected strong cell-cell interaction component (see summary 3.2.1), a gene overexpression assay was performed for biological conformation. In summary, again as expected (summary 3.2), the authors conclude from

these results that intracellular expressions 'are closely related to the cellular states of neighbors and the spatial location of cells in the global environment'.

The selection of markers for this experiment clearly biases the modeling outcome. More than half of the proteins are mitotic or cell-cycle related, therefore a linear model actually would explain 'on average half of the variability' (see abstract), when cells enter specific cell cycle states, i.e. these observed pathways are coordinated.

We agree that the chosen marker panel will to some extent affect the model outcome, and we have added an acknowledgement of this to the Discussion. The aim of this study was to examine cellular heterogeneity and the factors influencing it, as well as phenotypic adaptation, in a simplified system. We expected that cells in spheroids would be likely to exhibit different growth phenotypes and exhibit spatial variation in growth signaling and we chose markers that could capture this variation. We do not claim to have comprehensively described phenotypic variation even in this simplified system, or to have identified all of the factors influencing it. Our marker panel was sufficient, however, to uncover the lack of independence of internal and environmental factors on marker variation.

We note that cell cycle is not the only determinant of marker variation. This can be clearly seen in Figure. 4A where cell cycle markers (yellow) on average only explain about 20% of total marker variability, with on average approximately 30% of marker variability is explained by non-cell cycle factors.

How useful or artificial are these 'modules' of intracellular, cellular, local, global in spheroids at all, if they are closely related?

We demonstrate in our paper that these factors are inter-related. We think that this demonstration is useful in that it shows that such factors should not be modeled independently, as in fact has been done in previous studies (SVCA, MISTY (Arnol *et al*, 2019; Tanevski *et al*). In the revised version, we now provide specific examples (for the markers carbonic anhydrase, pS6 and pRb) of the implication of modeling such factors independently. We think this is an important finding for the systems biology community as it affects how such spatial data should be modeled.

And what do we learn from these comparisons to explain e.g. cancer (see motivation in introduction)?

The aim of this study was not to explain or even study cancer. We discussed cancer in the Introduction because there is at present great interest in generating spatial atlases of tumour tissue. In such studies, multiplexed imaging is employed with the goal of identifying drivers of tumor heterogeneity and interactions in the tumor microenvironment (Jackson *et al.*, 2020; Ali *et al.*, 2020; Moffitt *et al.*, 2018; Shah *et al.*, 2017; Regev *et al.*, 2017). The point we sought to make in the Introduction is that such datasets – although definitely highly relevant for biomedical research – present challenges for the investigation of phenotypic plasticity, as the problem of phenotypic variability is convoluted with the existence of multiple cell types, lineages, unknown environmental gradients, cell-type co-localization due to structured tissues, and likely other factors. Insights gained from our work on highly simplified systems

will inform spatial analyses of phenotypic variation in more complex, more biologically relevant systems such as tumour tissue. We had included some discussion of this point in the initial version, and in the Discussion section of the revision we address it in more detail.

Especially, knowing that cancer does not evolve monoclonal, nor cell lines are monogenic (<https://dmm.biologists.org/content/11/11/dmm037366>)? Cell lines are polyclonal and are composed of different cell states and types, which means that derived multicellular tumorspheroids are by definition heterogenous. The current manuscript does not introduce, model nor discuss this aspect.

We have not examined the genomes of our cell lines and therefore cannot rule out polyclonality. Even if these cell lines show a certain level of polyclonality, then this does not automatically mean that this will affect cell phenotype and state. Please note that a major advantage of our multiplexed imaging approach is the comprehensive analysis of heterogeneous cell phenotypes and states. Given that i) we did not observe vast phenotypic heterogeneity in the studied cell lines and ii) these lines reproducibly grew into stereotypic structures where phenotypic variation is strongly dominated by the environment (i.e., these cells form radially symmetric spheres), we do not believe that we are dealing with high levels of genotypic cell-to-cell variability.

Therefore, above 'findings' might be not valid anymore, if tumor cell type 'mixtures' would be used with strong inter-spheroid heterogeneity (also in time).

Response: This is a valid point that we address in our Discussion section “Challenges in adapting the analysis to complex tissues”. As mentioned above, we have expanded this section in the revised version.

In the context of known and shown inter-spheroid heterogeneity (referring to Fig. 2a, example IMC images of a spheroid plug - HT29 only -, top row), the manuscript is not informative or 'tangible'.

There may be a misunderstanding. We observe very little inter-spheroid heterogeneity for a given cell line. The heterogeneity observed in the top row of Figure 2a is because each spheroid cut contains spheres from all four cell lines. We have made this clear in the revised manuscript.

The assumption of basic understanding disease phenotypic patterns within spheroids by established cell lines might be not given, because malignant or patient-derived material would display different scenarios using more appropriate marker panels (e.g. epigenetic/developmental).

The aim of this study was to use a multiplexed readout in a very simplified model system to examine cellular heterogeneity and the factors that influence it. Although we do emphasize that the systems-level interdependencies we describe here should be considered when studying more complex systems, we do not claim to understand disease phenotypic patterns, and we explicitly state that doing similar studies in more heterogeneous systems

will be very challenging. We have added text to the Discussion section in the hope of making this clearer.

Fig.2c should have a clear separation by headers. Not sure if Fig. 1a and Fig. 3a, c, d, e schemes are helpful for understanding of the points/models made/computed.

We have tried to separate the headers more clearly in Figure 2c in the revised version. Figures 1a, 3a, 3c-e are graphical models that have a specific meaning to the statistical community. We have reorganized Figure 1a, and annotated all these models to clarify for non-experts.

Despite the excellent technical and analytical methods presented in the manuscript, the drawn interpretations are not very biological insightful or novel, while the fundamental question of biological heterogeneity and wired interplay cannot be addressed due to the experimental design. In conclusion, the technical quality would justify publication of the manuscript, while biological application/findings and relevance might not.

As stated earlier, one of the major conclusions of the manuscript is that the interdependency of factors influencing cellular phenotype (i.e., marker expression) in 3D tissues should be considered in order to accurately describe the sources of cellular heterogeneity. It is likely that the specific influences will differ in more physiological systems like tumour tissues, but we would expect that the interdependency of factors influencing cell phenotype would still apply. We trust that we have made this clear in the revised manuscript.

References

- Arnol D, Schapiro D, Bodenmiller B, Saez-Rodriguez J & Stegle O (2019) Modeling Cell-Cell Interactions from Spatial Molecular Data with Spatial Variance Component Analysis. *Cell Rep* 29: 202–211.e6
- Catena R, Özcan A, Kütt L, Plüss A, IMAXT Consortium, Schraml P, Moch H & Bodenmiller B (2020) Highly multiplexed molecular and cellular mapping of breast cancer tissue in three dimensions using mass tomography. 2020.05.24.113571
- Durand RE (1982) Use of Hoechst 33342 for cell selection from multicell systems. *J Histochem Cytochem* 30: 117–122
- Tanevski J, Gabor A, Flores ROR, Schapiro D & Saez-Rodriguez J Explainable multi-view framework for dissecting inter-cellular signaling from highly multiplexed spatial data. doi:10.1101/2020.05.08.084145 [PREPRINT]

Thank you for sending us your revised manuscript. We think that the performed revisions satisfactorily address the issues raised by the reviewers. I am glad to inform you that we can soon formally accept your manuscript for publication, pending some editorial issues listed below.

2nd Authors' Response to Reviewers**11th Nov 2020**

The authors have made all of the requested editorial changes.

Accepted**12th Nov 2020**

Thank you again for sending us your revised manuscript. We are now satisfied with the modifications made and I am pleased to inform you that your paper has been accepted for publication.

*

Corresponding Author Name: Bernd Bodenmiller
Journal Submitted to: Molecular Systems Biology
Manuscript Number: - MSB-20-9798